# In situ atomic-scale observation of dislocation climb and grain boundary evolution in nanostructured metal

Shufen Chu[1], Pan Liu [1,2✉], Yin Zhang[3], Xiaodong Wang[1], Shuangxi Song [1], Ting Zhu [3✉], Ze Zhang[4], Xiaodong Han [5], Baode Sun[1] & Mingwei Chen [6✉]

Non-conservative dislocation climb plays a unique role in the plastic deformation and creep of crystalline materials. Nevertheless, the underlying atomic-scale mechanisms of dislocation climb have not been explored by direct experimental observations. Here, we report atomic-scale observations of grain boundary (GB) dislocation climb in nanostructured Au during in situ straining at room temperature. The climb of a edge dislocation is found to occur by stress-induced reconstruction of two neighboring atomic columns at the edge of an extra half atomic plane in the dislocation core. This is different from the conventional belief of dislocation climb by destruction or construction of a single atomic column at the dislocation core. The atomic route of the dislocation climb we proposed is demonstrated to be energetically favorable by Monte Carlo simulations. Our in situ observations also reveal GB evolution through dislocation climb at room temperature, which suggests a means of controlling microstructures and properties of nanostructured metals.

[1] Shanghai Key Laboratory of Advanced High-temperature Materials and Precision Forming, State Key Laboratory of Metal Matrix Composites, School of Materials Science and Engineering, Shanghai Jiao Tong University, Shanghai, China. [2] WPI Advanced Institute for Materials Research, Tohoku University, Sendai, Japan. [3] Woodruff School of Mechanical Engineering, Georgia Institute of Technology, Atlanta, GA, USA. [4] Center of Electron Microscopy and State Key Laboratory of Silicon Materials, School of Materials Science and Engineering, Zhejiang University, Hangzhou, China. [5] Institute of Microstructure and Properties of Advanced Materials, Beijing University of Technology, Beijing, China. [6] Department of Materials Science and Engineering, Johns Hopkins University, Baltimore, MD, USA. ✉email: panliu@sjtu.edu.cn; ting.zhu@me.gatech.edu; mwchen@jhu.edu

Nanostructured metals often exhibit superior mechanical properties due to strong size effects at the nanoscale[1,2]. They have been used as important components for modern nanotechnology and electronics[3]. In general, the nanoscale size of crystals affects the generation, motion, reaction, and annihilation of dislocations[4–10]. However, the atomic-scale mechanisms of dislocation movement in nanosized structures remain elusive, particularly for climb of dislocations that involves the non-conservative motion out of original slip planes. Theoretical calculations and atomistic simulations have been performed to investigate dislocation climb over relevant time and temperature scales[11–15]. Mesoscale simulations based on discrete dislocation dynamics[16–20] and phase field methods[21–24] have been also used to investigate dislocation climb. However, in situ atomic-scale experimental observations of dislocation climb have not been realized to our knowledge. Hence, the dynamic evolution of dislocation cores during their climb processes is not clearly understood, and the nanoscale size effects on dislocation climb remain little known.

Here, we report in situ atomic-scale observations of dislocation climb at a tilt grain boundary (GB) in an Au ligament within de-alloyed nanoporous gold with the face-centered cubic (FCC) structure[25]. Climb of GB dislocations is driven by applied bending load at room temperature. As a result, most of climbing GB dislocations escape from the free surface of the ligament, leading to the evolution of a high-angle GB (HAGB) to a Σ3 twin boundary (TB). The observed atomic-scale climbing processes of dislocations unveil a unexpected route of non-conservative dislocation motion at room temperature, which is different from conventional understanding.

## Results

**Dislocation climb and GB evolution.** In situ straining experiment was conducted using a double-tilt actuator inside a transmission electron microscope (TEM)[26]. High-resolution TEM (HRTEM) images were recorded at a high rate of 30 frames per second, which greatly facilitated the tracking of step-by-step movements of individual GB dislocations. Figure 1a shows the atomic structure of an Au ligament of diameter ~7 nm before

loading (time $t = 0$ s). The reconstructed atomic model of the ligament is shown in Fig. 1h. The ligament consists of three nano-sized grains, with a Σ3 TB between the left and middle grains and an HAGB between the middle and right grains. All three grains are aligned along the [$\bar{1}$10] zone axis. Simulated atomic images indicate that dark spots in the HRTEM images correspond to Au atomic columns (Supplementary Fig. 1). Details of HRTEM image simulations are provided in Supplementary Note 1 and Supplementary Table 1. The HAGB consists of a TB and a nearby array of edge dislocations aligned vertically above each other. Since these dislocations are close to, but away from the TB, the edge components of the Burgers vectors can be clearly determined from HRTEM images by Burgers circuit analyses (inset of Fig. 1a). Combined with an atomistic model reconstructed from the HRTEM image, the full Burgers vectors of these dislocations can be identified as $1/2[011](11\bar{1})$ or $1/2[101](11\bar{1})$ (Supplementary Fig. 2). The core of each dislocation is marked by a ⊥ symbol placed right below the extra half-plane in Fig. 1a and by red atomic columns in Fig. 1h. The GB dislocations exhibit negligible core dissociation which may arise from the geometrical constraints from small dislocation spacings, grain sizes, and large applied stresses, despite a low stacking fault energy of Au (32–50 mJ m$^{-2}$ [27,28]). They are neither the non-dissociated nor dissociated 1/3 <111> TB disconnections[29–31]. Each GB dislocation in Fig. 1a is assigned with a number between "1" and "9" to facilitate the tracking of their movements. These dislocations distribute uniformly with an average spacing of 0.68 nm. The angle θ between the respective (111) plane in the middle and right grains (i.e., (111)$_L$ and (111)$_R$ in Fig. 1a) is measured to track the change of misorientation between the two grains, and it is decreased to zero when both (111)$_L$ and (111)$_R$ planes become the (111) TB plane. During in situ testing, the Au ligament was subjected to an effective bending moment, as indicated in Fig. 1h. As a result, non-uniform normal stresses were exerted on the HAGB aligned approximately with the cross section of the ligament, and they will be shown later by strain mapping through geometric phase analysis (GPA)[32].

The HRTEM images in Fig. 1b–g reveal the dynamic rearrangement of dislocations "1" and "9" at the HAGB. Detailed

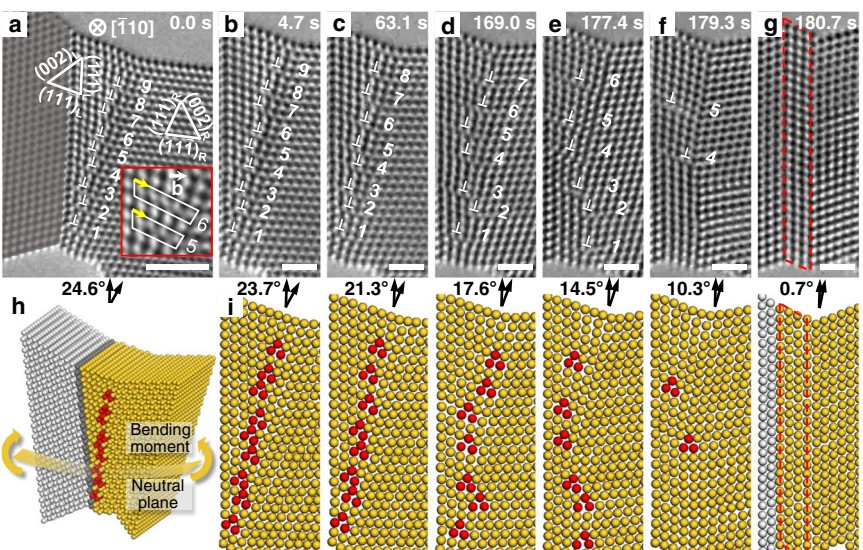

**Fig. 1 Representative in situ HRTEM images showing room-temperature dislocation climb at an HAGB and atomic models reconstructed from the HRTEM images. a–g** Dislocations are marked by both symbols ⊥ and numbers. Burgers circuits in the inset of **a** indicate the identical edge components of the dislocations. The newly formed twin lamella is denoted by red dashed lines in **g**. Scale bar: **a** 2 nm and **b–g** 1 nm. **h** Atomic model reconstructed from the HRTEM image in **a**. The pre-existing Σ3 TB is marked by a dark grey plane and the dislocation cores are highlighted by red atomic columns. The yellow curved plane indicates the neutral plane of bending. **i** Reconstructed atomic configurations corresponding to the HRTEM images in **b–g**.

GB dislocation movements between these HRTEM images can be seen in Supplementary Fig. 3–5 and Movie 1. In Fig. 1b–g, the left TB is not included for clarity, as it remains nearly intact throughout in situ testing. The atomic configurations reconstructed from these HRTEM images are shown in Fig. 1i, where the $\theta$ value is given for each frame. During in situ testing, $\theta$ decreased gradually, giving a total rotation of ~24° between the adjoining grains across the HAGB. The relationship between $\theta$ and the total dislocation spacing at the GB obeys the Frank's equation[33] (see Supplementary Fig. 6 and Supplementary Note 2). From 0 s (Fig. 1a) to 4.7 s (Fig. 1b), the upmost dislocation "9" moved up one atomic spacing out of its original {111} slip plane. This non-conservative dislocation motion, called positive climb, resulted in removal of one atomic column at the dislocation core, causing a decrease of $\theta$ to 23.7°. From 4.7 s to 63.1 s, positive climb occurred for dislocations "7" to "9" (Fig. 1c and Supplementary Fig. 3). These climb motions resulted in a further decrease of $\theta$ to 21.3° and also led to the annihilation of dislocation "9" from the top surface of the ligament. We usually observed the climb of a single dislocation at a time, while other neighboring dislocations remained stationary. From 63.1 s to 169.0 s (Fig. 1d), positive climb occurred for dislocations "4" to "8", leading to annihilation of "8" from the top surface of the ligament. These GB dislocations also glided occasionally along the horizontal {111} slip planes (Supplementary Fig. 3-4), so that the vertical alignment of GB dislocations in Fig. 1d changed compared with that in Fig. 1c. From 169.0 s to 177.4 s (Fig. 1e), dislocations "6" and "7" further climbed upward, leading to annihilation of dislocation "7" from the top surface of the ligament. As a result, the GB became close to a TB with a sharp interface in Fig. 1e. During this interval, dislocation "1" moved down one atomic spacing out of its original {111} slip plane, called negative climb. From 177.4 s to 179.3 s (Fig. 1f), only dislocations "5" and "4" were left at the GB. The detailed annihilation process of dislocation "6" was not captured. This dislocation likely underwent positive climb and escaped the ligament from its top surface, similar to dislocations "7" to "9". Note that dislocations "1" to "3" glided along the horizontal {111} planes in the middle grain, transmitted across the GB, and further glided into the interior of the right grain (Supplementary Fig. 3–5 and Supplementary Fig. 7). As a result, the local TB migrated to the right. At 180.7 s (Fig. 1g), dislocations "5" and "4" disappeared. Successive HRTEM images in Supplementary Fig. 8a–d indicate that dislocations "5" and "4" underwent positive climb and escaped the ligament from its top surface. After the entire array of 9 GB dislocations disappeared, the HAGB at 0 s became a coherent TB at 180.7 s, with the concomitant formation of a thin twin lamella of four atomic layers thick between the newly formed TB and the preexisting TB (Fig. 1g and Supplementary Fig. 5).

During the above in situ straining experiment, climb of GB dislocations occurred predominantly, resulting in plastic bending deformation of the ligament with a drastic decrease of grain misorientation by ~24°. The non-conservative movements of GB dislocations were driven by the bending-induced normal stresses on the HAGB plane, being compressive in the upper part and tensile in the lower part of the ligament, as shown by the GPA maps in Supplementary Fig. 9. The details of the GPA analysis are provided in Supplementary Note 3 and Supplementary Fig. 10–11. The positive climb of GB dislocations removed atomic columns at the dislocation cores so as to accommodate the compressive stresses on the HAGB plane, while the negative climb of GB dislocations introduced atomic columns to accommodate the tensile stresses. The local normal stress is estimated to be ~3.2 GPa based on the corresponding maximum lattice strain of ~4% indicated by the strain profile in Supplementary Fig. 9f and

Young's modulus of ~80 GPa for Au. The large normal stress, together with constraints from GBs, leads to the negligible dissociation of the 1/2 < 011 > dislocations despite the low stacking fault energy of Au. The neutral plane with zero local strain is closer to the bottom surface of the ligament. Hence, a major portion of the HAGB (i.e., above the neutral plane) was subjected to compressive stresses, driving the positive climb of dislocations "4" to "9". Dislocations "6" to "9" climbed before dislocations "4" and "5", because they were at larger distances from the neutral plane and thus under higher compressive stresses for driving their climb motions. In contrast, dislocations "1" to "3" were located below/close to the neutral plane and thus under relatively low tensile stresses. Hence, they exhibited the conservative motion of glide into the right grain, with occasional negative climb to accommodate local deformation incompatibility. During these processes, slip transmission of dislocations "1" to "3" across the GB led to local GB migration. Such dislocation glide activities indicate the buildup of large local shear stresses. Overall, the applied bending load resulted in the evolution of an HAGB to a TB during in situ straining primarily through dislocation climb and annihilation to the top surface above the neutral plane, in conjunction with dislocation glide and transmission across the GB below the neutral plane.

**Climb velocities measurement**. The high frame rate of HRTEM imaging enabled us to measure the climb velocities for dislocations "1" to "9". The maximum climb velocities of individual dislocations are given in Supplementary Table 2. As described earlier, the positive climb of dislocations "6" to "9" occurred before dislocations "4" and "5". The maximum climb velocities of the former group are lower than the latter. Such different velocities can be rationalized using the aforementioned Frank's equation. Assuming an approximately constant rate of change of $\theta$ during bending, the climb velocity should increase with increasing spacing between GB dislocations. As dislocations "6" to "9" left the ligament, the dislocation spacings were increased, resulting in higher climb velocities of dislocations "4" and "5". In contrast to the positive climb of dislocations "4" to "9", dislocations "2" and "3" were close to the neutral plane and subjected to low tensile stresses. As a result, their driving forces of climb were low, giving near-zero climb velocities. Dislocation "1" was below but further away from the neutral plane, thus exhibiting occasional negative climb. To gain a quantitative sense of the climb velocities, we note that the core of dislocation "4" moved up ~0.80 nm within ~0.5 s (Fig. 2 and Supplementary Movie 2). Figure 2f shows the corresponding displacement and velocity of dislocation "4" against time measured from the HRTEM images in Fig. 2a–e. Despite a late start of climb, dislocation "4" exhibited a high maximum climb velocity of 17.2 nm s$^{-1}$ and vanished at the top surface of the ligament within ~1 s (Supplementary Fig. 5c–e).

**Reconstruction of dislocation core during positive climb process**. The above results demonstrate the predominant dislocation climb in a nanostructured metal at room temperature that plays an important role in bending deformation and the stress-induced GB evolution. Although several models have been proposed on the atomistic mechanisms of dislocation climb[14,15,34–38], the real atomic processes have not been observed through in situ experiments. Our atomic-scale in situ observations (Figs. 3–4 and Supplementary Movie 3–4) captured the step-by-step movements of dislocation climb and particularly the corresponding destruction or construction process of an atomic column at the dislocation core. As an example, Fig. 3a–f shows in situ HRTEM images of the positive climb of dislocation "7" by moving up one

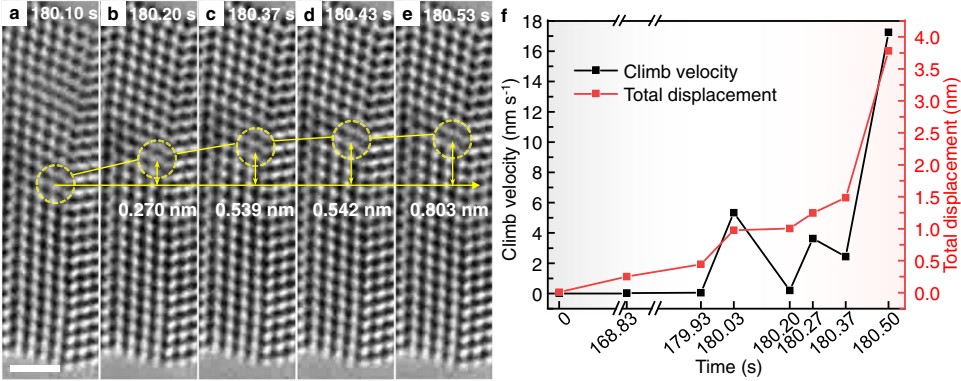

**Fig. 2 Climb velocity and displacement of dislocation "4" measured from the HRTEM images. a-e** HRTEM images showing fast movement of the core (circled by dashed lines). Scale bar: 1 nm. **f** Climb velocity and displacement of dislocation "4" as a function of time during the whole in situ straining process.

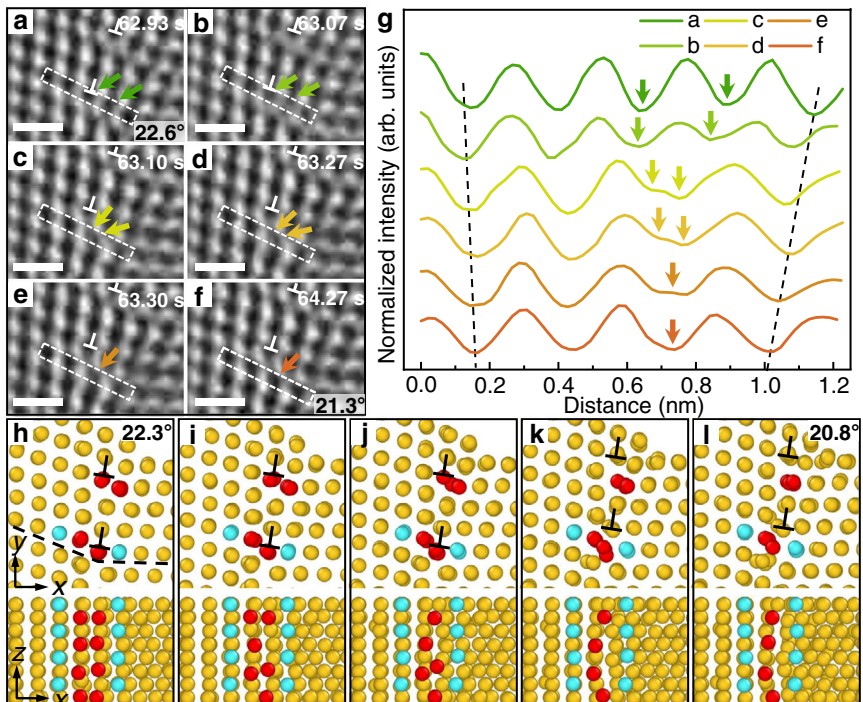

**Fig. 3 Reconstruction of dislocation cores during positive climb. a–f** A series of HRTEM images showing the reconstruction of a dislocation core. Scale bar: 0.5 nm. **g** The corresponding contrast intensity profiles extracted along the dotted rectangles in **a–f**. **h–l** GCMC results revealing the atomic structure evolution of dislocation cores during positive climb. The upper row of images shows the top view (xy-plane) of the merging process of two adjacent atomic columns (colored in red) at the core of each of two GB dislocations (marked by symbol ⊥). The lower row shows the corresponding side view (xz-plane) generated by cutting the atomic structure along the black dotted line in **h**. Another two neighboring atomic columns (colored in cyan) serve as a reference and are not directly involved in the reconstruction process at the climbing dislocation core.

atomic spacing out of its original {111} slip plane. Figure 3g shows the corresponding line profiles of normalized intensity across the dislocation core, as extracted from the rectangles enclosing a {111} atomic layer in Fig. 3a–f. While the interpretation of HRTEM contrast and associated line profiles can be difficult, the small sample thickness and minimized spherical aberration make the interpretation more straightforward on the basis of a charge density projection approximation[39] (see Supplementary Note 4). By directly correlating the number of atoms in a column with contrast intensity, a semi-quantitative analysis of the HRTEM images can be realized. Importantly, the changes of intensity at the atomic columns (i.e., valleys) in Fig. 3g indicate

that the climbing process of a GB dislocation involves the reconstruction of two adjacent atomic columns at the dislocation core, rather than a single atomic column at the end of the extra half-plane. When a climb event began, the intensities at the two valleys in the profile *b*, as marked by green arrows, increased significantly in comparison with the profile *a*. As this GB dislocation continued to climb, the two valleys in the profiles *a* and *b* gradually merged into one valley in the profiles *c* to *f*, indicating the two atomic columns merge into one at the dislocation core. Note that the location of the intensity peak between the two valleys in the profiles *a* and *b* was occupied by the newly formed valley in the profile *f*, while its neighboring valleys also adjusted their positions

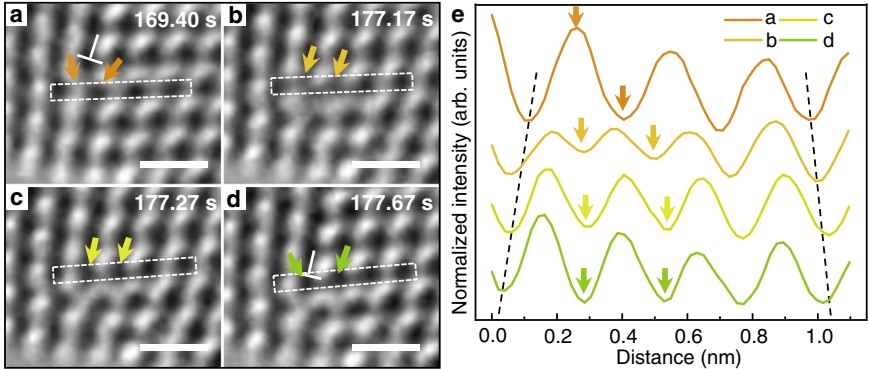

**Fig. 4 Reconstruction of a dislocation core during negative climb. a–d** A series of HRTEM images showing the reconstruction process. Scale bar: 0.5 nm.
**e** The corresponding contrast intensity profiles extracted along the dotted rectangles in **a–d**.

to accommodate these changes. Hence, 5 atomic columns in the profile *a* transformed into 4 columns in the profile *f*, as the dislocation core moved up one atomic layer out of its original {111} plane. This transformation reflects a drastic local lattice contraction, as marked with two black dashed lines in Fig. 3g, and thus accommodates the local compressive stress arising from the applied bending load. Simultaneous climb of dislocation "7" and dislocation "8" caused a decrease of θ from 22.6° to 21.3°, as seen in Fig. 3a, f. Additionally, the HRTEM images are processed with false color to enhance contrast variation at the dislocation cores (Supplementary Fig. 12). The same atomic processes at the cores of positively climbing dislocations were observed (Supplementary Fig. 13 and Movie 5). Based on the in situ observations in Fig. 3, Supplementary Fig. 14 shows the reconstructed atomic configurations at the core of a climbing dislocation along with simulated HRTEM images which give the same profile evolution as Fig. 3g. In addition, the intensity profiles of atomic columns were analyzed by treating bright spots (i.e., peaks) in the HRTEM images as atomic columns, giving the equivalent result of reconstruction of the core of a climbing GB dislocation (Supplementary Fig. 15).

To understand the atomic processes at the core of climbing GB dislocations, we performed Grand Canonical Monte Carlo (GCMC) simulations using an atomic model of Au ligament reconstructed from the HRTEM image. As shown in Supplementary Fig. 16, the left end of the reconstructed ligament was fixed and a bending load was applied to the right end. Figure 3h–l displays the top views (first row of images) and side views (second row) of GCMC results in a GB region containing two neighboring dislocation cores (black boxed region in Supplementary Fig. 16), which experienced compressive stresses from the bending load. In the top views, two atomic columns at the core of each GB dislocation were directly involved in the simulated climbing process and thus colored in red; two neighboring atomic columns were colored in cyan to serve as a reference. Since GCMC simulations allowed the removal and insertion of atoms to lower the system energy, local chemomechanical equilibrium was maintained at the core of the climbing dislocation. At the beginning of dislocation climb, the compressive stress led to local lattice distortion with decreased distance between the two atomic columns in red (Fig. 3i), driving atoms in these two columns to move toward each other (Fig. 3j). Eventually, the two atomic columns merged into one column (Fig. 3k–l), resulting in the positive dislocation climb by one atomic layer. Meanwhile, the θ value decreased from 22.3° (Fig. 3h) to 20.8° (Fig. 3l), which is comparable with our experimental results (Fig. 3a–f). As seen from the corresponding side views, merging of the two atomic columns at each dislocation core involved a series of unit processes of removing one atom from a pair of atoms along the direction of sample thickness, while the two adjacent atomic

columns in cyan were not involved. These GCMC results directly support our in situ HRTEM observations of positive climb of each dislocation core through merging of two atomic columns as an energetically favorable process. Moreover, the GCMC also reveals the highly localized processes of merging of atom pairs along the direction of sample thickness, which cannot be observed from the HRTEM images that only show the projection of all atomic layers along the [1̄10] viewing direction. Apparently, to experimentally resolve the individual atom motion in climbing dislocation cores, a new TEM technique, such as such as atomic resolution electron tomography[40], is required in future studies.

**Reconstruction of dislocation core during negative climb process.** Our HRTEM observations also capture the atomic process of negative climb of GB dislocations, which involves the insertion of an atomic column at the dislocation core. Figure 4a–d shows in situ HRTEM images of the negative climb of dislocation "1". Figure 4e shows the corresponding line profiles of normalized intensity across the dislocation core, as extracted from the respective rectangle enclosing a {111} atomic layer in Fig. 4a–d. When negative climb began, the intensity valley indicated by the right arrow in the profile *a* increased dramatically in the profile *b*. Meanwhile, the intensity peak indicated by the left arrow in the profile *a* gradually became a valley in the line profile *d*, corresponding to the formation of a new atomic column. This indicates the splitting of a single atomic column into two columns by atom diffusion to the edge of the extra half-plane. During this process, other neighboring valleys were shifted to accommodate the newly inserted valley, i.e., atomic column. Hence, 4 atomic columns (valleys) in the profile *a* transformed into 5 columns (valleys) in the profile *e*, as the dislocation core moved down one atomic layer from its original {111} slip plane. This transformation reflected a drastic local lattice expansion, as marked with two dashed lines in Fig. 4e, and thus accommodated the local tensile stress arising from the bending load. Negative climb of GB dislocations was also observed in our GCMC simulations, as shown in Supplementary Fig. 17. The combined in situ HRTEM and GCMC results provide a solid basis for future studies of kinetic pathways of atom diffusion at the core of climbing dislocations. It should be noted that the cores of GB dislocations in our work exhibit negligible dissociation. Such compact core structures facilitate the observed climbing processes. The climb of GB dislocations can become more complicated when they dissociate into partial dislocations and thus warrants further in situ study in the future.

## Discussion

By tracking the intensity profile at a dislocation core during its climbing process, together with GCMC simulations, our results

reveal the atomic processes of stress-driven climb of GB dis-locations at room temperature. In contrast to the conventional model of dislocation climb by removal or insertion of a single atomic column at the edge of the extra half atomic plane, we find the merging of two atomic columns into one atomic column for positive climb and the splitting of one atomic column into two atomic columns for negative climb. This route could be more energetically favorable compared to diffusion along one atomic column, as supported by our GCMC simulations. Our conclusion is also applicable to the cases where the boundaries conditions are symmetric, as evidenced by further GCMC simulations (Supple-mentary Fig. 18). These volume non-conservative processes are driven by the normal stress acting on the local GB plane. They would involve the diffusion of vacancies/atoms to/away from the jog pairs at the core of a dislocation line, and warrant further study in the future. In our work, dislocation climb at room temperature is mainly driven by applied high stresses enabled by nanosized structures. The electron beam effect on the sample temperate increase was estimated to be less than 5 K (see Sup-plementary Note 5). Radiolysis to the Au specimen from inelastic scattering of the electron beam is significantly suppressed[41]. Unlike dislocation depinning caused by a displacement cascade effect under ion irradiation[42], a dislocation cascade induced by electron irradiation is not expected in an Au specimen with high knock-on and sputtering energies[43]. In other words, at an accelerating voltage of 200 keV, knock-on displacements induced by the electron beam should have a negligible effect on the Au specimen (see Supplementary Note 6). Therefore, we conclude that the massive vacancy-atom exchange required for dislocation climb in the sample was driven predominantly by high local stresses rather than electron beam irradiation. According to our GPA results and GCMC simulations, the local normal stress on the HAGB is estimated to be ~3.2 GPa using Young's modulus of ~80 GPa for Au. In addition, in the absence of a native oxide layer on the Au specimen, electron beam-enhanced surface dislocation nucleation was not observed[44]. The stability of GBs under pro-longed electron beam irradiation was evident (Supplementary Fig. 19). It has been reported that electron beam irradiation can accelerate a deformation mechanism rather than change to a different mechanism[45]. Hence, electron beam-assisted dislocation activation is possible[43] and may facilitate the climb and slip behavior of dislocations. When the bending load is applied to the Au ligament, the sub-10 nm size facilitates dislocation climb at room temperature because the free surface nearby can act as a highly efficient vacancy source. Although the pre-existing TB may also facilitate the room temperature dislocation climb by provid-ing fast diffusion paths, it does not appear to play an essential role in the observed dislocation climb processes since lattice disloca-tion climb was observed directly (Supplementary Fig. 8e–h).

In summary, our in situ TEM observations reveal the atomic processes of dislocation climb in an FCC metal. Dislocation climb occurs by stress-induced reconstruction of two atomic columns at the edge of an extra half atomic plane in the dislocation core, which contrasts with the conventional single atomic column models. The finding of dislocation climb at room temperature may offer new insights into the unique time-dependent properties of nanocrystalline materials, such as room-temperature creep, grain coarsening, and superplasticity[1,46–49]. In the current pre-vailing models, these low-temperature time-dependent properties are usually attributed to fast diffusion along GBs in nano-structured materials. The dislocation climb induced GB evolution, as observed in our study, provides a mechanism of fast GB deformation and evolution. Our results also suggest a potential route to develop strong and ductile nanostructured metals by controlling dislocation climb through dislocation pinning by chemical doping or by promoting dislocation climb through

introducing vacancies to tailor their strength and deformability at room temperature.

## Methods

**Preparation of nanoporous gold (NPG) film**. An NPG film of 100 nm thickness was prepared by electrochemical dealloying of Ag from an Ag$_{65}$Au$_{35}$ (at%) white Au leaf in a 70 vol% HNO$_3$ solution, followed by carefully rinsed with distilled water to remove the residual nitric acid. After a few times washing, the NPG film was cut into pieces by a knife blade. The actual size of NPG thin films ranges between 50 μm to 1 mm. Under an optical microscope, we utilized the tweezer to clamp the strain actuator and scooped out the NPG film from the deionized water to transfer it to the straining actuator. Before being inserted into the TEM chamber, the holder with the loaded NPG film sample was processed by vacuum drying and plasma cleaner to remove possible organic contamination. The as-prepared NPG film has three-dimensional nanopore channels and Au ligaments. The average size of Au ligaments is about 20 nm with a wide distribution range from ~5 to 35 nm. The Au ligaments have an average chemical composition of Au$_{97}$Ag$_3$ (at%), measured by TEM equipped with an energy dispersive X-ray spectrometer. The NPG sample was used as a model system to study the mechanism of dislocation climb in FCC nanostructured metals. Many GBs are available in the ligaments of as-prepared NPG films. Thus, it is not difficult to identify an HAGB for in situ straining experiment.

**In situ TEM straining experiment**. In situ bending test was carried out using a double-tilt straining holder (Bestron Science and Technology Co., LTD, Beijing). A miniature device was fixed on this custom-designed TEM sample holder to apply load by a strain actuator. The holder enables controllable bending, tension, compression deformation of the sample and a range of ±25° X-tilt and ±25° Y-tilt[26]. The NPG sample was subjected to a strain rate of ~10$^{-3}$ s$^{-1}$ by controlling the actuator; meanwhile, the NPG sample was tilted to a desired crystallographic zone axis for atomic-scale observation.

**Cs-corrected HRTEM characterization**. We used a 200 kV JEM ARM200F electron microscope (JEOL) equipped with a spherical aberration corrector (CEOS GmbH) for the objective lens system. With the optimized Cs correction value, the point-to-point resolution of the TEM images is better than 1.3 Å. The residual aberrations are as follows (95% certificate): two-fold astigmatism A$_1$ = 636.6 pm, three-fold astigmatism A$_2$ = 25.4 nm, axial coma B$_2$ = 13.1 nm, spherical aberra-tion coefficient Cs = −1.15 μm, four-fold astigmatism A$_3$ = 1.1 μm, star aberration S$_3$ = 1.3 μm, five-fold astigmatism A$_4$ = 29.4 μm. Atomic-scale deformation pro-cesses were recorded by a Gatan charge-coupled device (CCD) camera.

**Grand canonical Monte Carlo simulations**. We performed atomistic Monte Carlo simulations to investigate the mechanisms of dislocation climb and GB evolution. The simulations were performed using LAMMPS[50] with an embedded-atom method potential of Au[51]. Based on the HRTEM image of the Au ligament in Fig. 1a, we set up the initial structure containing a Σ3 TB on the left and an HAGB on the right, as shown in Supplementary Fig. 16. Periodic boundary condition was applied in the out-of-plane direction of <110>. The initial structure was relaxed by the conjugate gradient method to reach a stress-free state. We used the Grand Canonical Monte Carlo (GCMC)[52] method to mimic the atomic diffusion processes at GB dislocations. This method allows us to perform conventional Monte Carlo moves and exchange GB atoms with a reservoir of atoms. To impose the bending load, we applied an incremental rotation to the right grain by 0.1 degree counter-clockwise and then performed 10000 GCMC steps at 300 K. The atoms in the green box were fixed to maintain the mechanical load. Due to the imposed rotation of the right grain, compressive and tensile stresses were generated on the top and bottom parts of the HAGB plane, respectively. In the GCMC simulation, atoms can be removed or inserted in the system based on the energy difference and the sampling temperature. We repeated the loading process by 50 incremental rotations with 10000 GCMC steps between every two consecutive rotations, in order to simulate the continuous increase of bending load in the experiment. When an atom column disappeared at the core of a GB dislocation, the positive/upward climb by one atomic layer was completed. Similarly, the negative/downward climb of a GB dislocation was simulated by the GCMC method, showing the insertion of an atomic column at the dislocation core. In the GCMC method, the chemical potential in the imaginary reservoir is −3.7 eV atom$^{-1}$, which is set to be 0.2 eV higher than the bulk cohesive energy to accelerate the simulated diffusion process.

## Data availability

The authors declare that the main data supporting the findings of this study are available within the article and its Supplementary Information files.

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

## Acknowledgements

P.L. is supported by Natural Science Foundation of China (52173224, 51821001, 52130105), Natural Science Foundation of Shanghai (21ZR1431200), and the Program for Professor of Special Appointment (Eastern Scholar) at Shanghai Institutions of Higher Learning. M.W.C. is sponsored by National Science Foundation (NSF-DMR-1804320).

## Author contributions

M.W.C. conceived and supervised the project. P.L. designed the experiments. S.F.C. and P.L. conducted the in situ Cs-corrected HRTEM experiments under the guidance of X.D.H. and Z.Z. Y.Z. and T.Z. performed the Monte Carlo simulations. S.F.C., P.L., X.D.W., S.X.S. and B.D.S. analyzed the data. S.F.C., P.L., T.Z. and M.W.C. wrote the manuscript. All authors contributed to the extensive discussions of the results.

## Competing interests

The authors declare no competing interests.
