## [Peer Review File · Nature Communications]

Title: In situ atomic-scale observation of dislocation climb and grain boundary evolution in nanostructured metalReviewers' comments:

Reviewer #1 (Remarks to the Author):

The authors have presented experimental observations of grain-boundary dislocation motion in a thin ligament of Au. They interpret this motion as climb and argue that the observation points to a new mechanism for dislocation climb and grain boundary reconstruction.

Unfortunately, the work is fundamentally flawed due to a misinterpretation of the defect configurations. As such, the analysis and interpretation of the significance of the work are not supportable. I also do not see novelty in the authors' claim that this work is pointing to a new route for dislocation climb.

Significantly, the authors have misinterpreted the observed grain boundary dislocations as $(1/2)\langle 110 \rangle$ crystal lattice dislocations. A key problem in the presentation is that the circuit analysis used to determine the Burgers vectors is invalid since it is being placed in context of a single crystal reference frame. From inspection, it is clear that the array of defects is separating two crystals that are close to a twin related orientation (inexplicably, the authors missed this point, despite overlaying Figure 1a with the traces of the Thompson's tetrahedron for the two adjacent crystal which can be seen to be rotated slightly away from the mirror twin configuration).

What the authors are actually observing here is a boundary that is vicinal to a $\Sigma 3$ twin with an array of $(1/3)\langle 111 \rangle$ twin boundary disconnections. The dislocation components of the disconnections are accommodating the angular rotation from the $\Sigma 3$ orientation and the step components are accommodating the deviation from the $\{111\}$ twin inclination. It is already known that such defects can move by climb and moreover that this climb despite can occur under irradiation conditions (e.g., in the electron microscope). It is not at all surprising that climb and removal of $(1/3)\langle 111 \rangle$ disconnections would eventually yield a twin boundary -- this is not a "reconstruction" of the boundary. Moreover, it is already known that $(1/3)\langle 111 \rangle$ disconnections, which often have a slightly dissociated core structure, can climb in this dissociated state (e.g., see work of Foiles et al.), so I don't see the novelty in the authors' argument that the observations are pointing to a phenomenon that differs from our existing understanding of climb.

The authors have also misrepresented and misapplied the Frank-Bilby equation which, as presented here, does not adequately account for the asymmetric inclination of the interface.

While the work presents some nice in situ observations, it is limited by its flawed analysis. Moreover, even if properly analyzed, I don't see that the present results provide sufficient novelty or fundamental new scientific insight to merit publication in Nature Communications.

Reviewer #2 (Remarks to the Author):

Although dislocation climb is a fundamental motion that plays critical roles in the mechanical and plastic properties of materials, knowledge of the atomic-scale mechanisms of dislocation climb remains limited. In this manuscript, the authors reported in situ atomic-scale observations of dislocation climb, which has not been reported previously in the literature. They found that climb of a grain boundary dislocation occurs by reconstruction of two atomic columns at the dislocation core. The new climb mechanism was validated by Monte Carlo simulations. This research is original. The manuscript is well-written. I recommend it for publication after the following comment is addressed.

The reported experiments depend on the free surfaces and applied bending load. Are the strengths of these point defect source and driving force comparable with those in the dislocation climb processes in a wide range of applications?

Reviewer #3 (Remarks to the Author):

The authors report in situ HRTEM observations of dislocation climb in a high angle grain boundary (HAGB) in nanoporous gold (np-Au). They find that dislocation climb involves rearrangement of two atomic columns rather than the destruction or construction of a single atomic column, as widely assumed in previous models. They have also performed Grand Canonical Monte Carlo (GCMC) simulations to understand the atomic processes involved in the dislocation climb. The GCMC simulations indicate that merging of two atomic columns into one (for positive climb) and splitting of a single atomic column into two (for negative climb) is energetically more favorable, which supports the in situ HRTEM observations. Although these results are interesting, the main conclusion of the paper, namely, that two atomic columns are involved in the climb process in FCC metals is somewhat premature.

Firstly, the authors assert that there is negligible core dissociation despite the low stacking fault energy of Au. While it is true that no apparent stacking faults are visible in the images, the resolution of the images is not sufficient to conclude that the dislocation core is restricted to a single atomic column. In fact, based on the intensity distribution in some of the HRTEM images (e.g., Supplementary Fig.6a) one might argue that the core is spread over two atomic columns. In that case, it is not surprising to see that two atomic columns are involved in the dislocation climb process.

Secondly, and more importantly, the authors do not consider the boundary conditions in interpreting the results. There is a σ_3 boundary on the left of the HAGB whereas there is no GB in the vicinity on the right. Therefore, the stress field near the dislocation core is not symmetric, which might lead to preferential diffusion in one direction and result in the involvement of two atomic columns. When the boundary conditions are symmetric, it is more plausible that three atomic columns (the dislocation core and one column to the left and right) rearrange to form two columns, which would also result in positive climb. While it might be difficult to find a HAGB flanked by two identical GBs on either side in experimental specimens, it should be relatively straightforward to simulate it using GCMC simulations. The authors should simulate this case and verify if only two atomic columns are still involved in the dislocation climb process.

Another aspect that the authors should consider is the possible role of the e-beam in activating the climb process. Although the authors argue that temperature increase due to e-beam exposure is minimal in Au, there are multiple reports of e-beam induced dislocation activation in nanostructured metals (including Au) even in the absence of significant temperature increase. Examples include

1. R. Sarkar, C. Rentenberger, J. Rajagopalan, Electron Beam Induced Artifacts During in situ TEM Deformation of Nanostructured Metals. *Scientific Reports*. 5, 16345 (2015).
2. M. Gaumé, P. Baldo, F. Momprou, F. Onimus, In-situ observation of an irradiation creep deformation mechanism in zirconium alloys. *Scripta Materialia*. 154, 87–91 (2018).
3. S.-H. Li, W.-Z. Han, Z.-W. Shan, Deformation of small-volume Al-4Cu alloy under electron beam irradiation. *Acta Materialia*. 141, 183–192 (2017).
4. S. Stangebye, Y. Zhang, S. Gupta, T. Zhu, O. Pierron, J. Kacher, Understanding and quantifying electron beam effects during in situ TEM nanomechanical tensile testing on metal thin films. *Acta Materialia*, 117441 (2021).

The authors should at least acknowledge this possibility and include the relevant references.

Point-by-point response to reviewers' comments (NCOMMS-21-29277)

We sincerely thank the reviewers for their careful reading of our manuscript and constructive comments on our work. In the following, our point-by-point response to each comment is highlighted in blue. We have revised the manuscript (highlighted in red) and supplementary materials accordingly.

Reviewer #1

The authors have presented experimental observations of grain-boundary dislocation motion in a thin ligament of Au. They interpret this motion as climb and argue that the observation points to a new mechanism for dislocation climb and grain boundary reconstruction.

Unfortunately, the work is fundamentally flawed due to a misinterpretation of the defect configurations. As such, the analysis and interpretation of the significance of the work are not supportable. I also do not see novelty in the authors' claim that this work is pointing to a new route for dislocation climb.

Significantly, the authors have misinterpreted the observed grain boundary dislocations as $(1/2)\langle 110 \rangle$ crystal lattice dislocations. A key problem in the presentation is that the circuit analysis used to determine the Burgers vectors is invalid since it is being placed in context of a single crystal reference frame. From inspection, it is clear that the array of defects is separating two crystals that are close to a twin related orientation (inexplicably, the authors missed this point, despite overlaying Figure 1a with the traces of the Thompson's tetrahedron for the two adjacent crystal which can be seen to be rotated slightly away from the mirror twin configuration).

What the authors are actually observing here is a boundary that is vicinal to a $\Sigma 3$ twin with an array of $(1/3)\langle 111 \rangle$ twin boundary disconnections. The dislocation components of the disconnections are accommodating the angular rotation from the $\Sigma 3$ orientation and the step components are accommodating the deviation from the $\{111\}$ twin inclination. It is already known that such defects can move by climb and moreover that this climb despite can occur under irradiation conditions (e.g., in the electron microscope). It is not at all surprising that climb and removal of $(1/3)\langle 111 \rangle$ disconnections would eventually yield a twin boundary -- this is not a "reconstruction" of the boundary. Moreover, it is already known that $(1/3)\langle 111 \rangle$ disconnections, which often have a slightly dissociated core structure, can climb in this dissociated state (e.g., see work of Foiles et al.), so I don't see the novelty in the authors' argument that the observations are pointing to a phenomenon that differs from our existing understanding of climb.

Response: We thank the reviewer for his/her comments on our analysis of grain boundary (GB) and dislocation types. Our analysis in the original submission is correct, while some discussions are not sufficiently clear and thus cause misunderstandings of the reviewer. In fact, we had recognized that the dislocation array studied is close to a twin boundary (TB). To avoid possible confusion in the revised manuscript, the high-angle GB studied is explicitly referred to as the combination of a

dislocation array and a TB. The predominant climb motions of dislocations in the array transform the high-angle GB to a pure TB. We emphasize that most dislocations in the array are close to, but not right on the TB plane during the GB transformation process. Importantly, the majority of these dislocations are not $1/3\langle 111 \rangle$ twin boundary disconnections, as discussed in detail below.

Fig. R1 (a) HRTEM image of a non-dissociated $1/3\langle 111 \rangle$ TB disconnection, after Marquis and Medlin¹. The TB step is shown by a relative shift between the $\{111\}$ TB planes (marked by black solid lines) on the two sides of the disconnection core (marked by a triangle). The extra half $\{111\}$ plane of the disconnection is marked by the pink dashed line and the symbol “ \perp ”. (b) HRTEM image of a dissociated $1/3\langle 111 \rangle$ TB disconnection, after Marquis and Medlin¹. This dissociated disconnection features a TB step and a stacking fault (marked by an inclined yellow solid line), along with an extra half $\{111\}$ plane (marked by the pink dashed line and the symbol “ \perp ”) away from the TB by several $\{111\}$ layers. (c-g) A series of representative HRTEM images in this work, showing that the TB (marked by the red line) remains atomically flat without TB steps during

dislocation climb. A clear Burgers circuit is shown in (f).

It is necessary to clarify what is the atomic structure of $1/3\langle 111 \rangle$ TB disconnections, which can involve the non-dissociated and dissociated types¹. First, let us consider the non-dissociated type of $1/3\langle 111 \rangle$ TB disconnections, each of which consists of a TB step as well as an extra half $\{111\}$ plane on the TB. **Fig. R1a** in this Response shows the HRTEM image of a non-dissociated $1/3\langle 111 \rangle$ TB disconnection, as taken from Fig. 2a in Marquis and Medlin¹. In **Fig. R1a**, the TB step is indicated by the two black lines with a relative shift by one $\{111\}$ layer, and the extra half $\{111\}$ plane (marked by the pink dashed line) is located immediately next to the TB plane. A similar HRTEM image of the $1/3\langle 111 \rangle$ TB disconnection can also be found in Fig. 7a of an earlier paper by Medlin et al.²

Next, let us consider the dissociated type of $1/3\langle 111 \rangle$ TB disconnections. **Fig. R1b** shows the HRTEM image of a dissociated $1/3\langle 111 \rangle$ TB disconnection, as taken from Fig. 2c in Marquis and Medlin¹. In this case, the dissociated TB disconnection can be considered to result from interaction between the TB and a dissociated full dislocation on the inclined $\{111\}$ plane (marked by the yellow solid line). That is, the leading partial of this dislocation is absorbed into the TB, resulting in a TB step indicated by two black lines with a relative shift in **Fig. R1b**. This TB step forms due to transformation of the leading partial into a stair-rod dislocation (see the corresponding schematic in Fig. 4 by Foiles and Medlin³). However, the trailing partial is outside the TB, as evidenced by the inclined stacking fault (marked by the yellow solid line) of a few atomic spacings wide. Moreover, the extra half $\{111\}$ plane (marked by the pink dashed line) is a few $\{111\}$ layers away from the TB. The core of the trailing partial is located at the intersection between the pink and yellow solid lines. From the above analysis, we note that the dissociated TB disconnection is characterized by a TB step, an inclined stacking fault, and an extra half $\{111\}$ plane outside the TB plane. Therefore, both **Fig. R1a** and **Fig. R1b** show that a $1/3\langle 111 \rangle$ TB disconnection, being either the non-dissociated or dissociated type, must be associated with a TB step.

In contrast, **Figs. R1c-g** show a series of representative HRTEM images in this work, where the TB (marked by the red line) remains atomically flat without TB steps during dislocation climb. In this set of images, we focus on the TB segment (marked by the red line) that can be tracked unambiguously. Several dislocations are located on the left side of this TB segment and they are inside in the grain rather than on the TB. Evidently, these climbing dislocations are neither the non-dissociated nor dissociated type of $1/3\langle 111 \rangle$ TB disconnections, because none of them is associated with a TB step.

Fig. R2a further shows an HRTEM image in this work containing a TB with steps as well as a dislocation array on the left side of this TB. These dislocations are not associated with steps on the TB either, and therefore are not $1/3\langle 111 \rangle$ TB disconnections. To understand this point, **Fig. R2b** shows the HRTEM image of a dissociated $1/3\langle 111 \rangle$ TB disconnection¹ (same as **Fig. R1b**). This image underscores the fact that the extra half $\{111\}$ plane (marked by the pink dashed line) of a dissociated $1/3\langle 111 \rangle$ TB disconnection must be located in the region enclosed by the inclined $\{111\}$ plane (marked by the yellow solid line) and the $\{111\}$ TB (marked by the black solid line) that form an **OBTUSE** angle ($\sim 110^\circ$) indicated by the red arrow. Moreover, Marquis and Medlin¹ showed

that the extra half $\{111\}$ plane of a dissociated $1/3\langle 111 \rangle$ TB disconnection **CANNOT** be located in the region enclosed by the inclined $\{111\}$ plane and the $\{111\}$ TB that form an **ACCUTE** angle ($\sim 70^\circ$) indicated by the green arrow in **Fig. R2b**. In fact, the extra half $\{111\}$ plane of each dislocation in **Fig. R2a** is located in the region enclosed by the corresponding inclined $\{111\}$ plane and the $\{111\}$ TB that form an **ACCUTE** angle ($\sim 70^\circ$). Therefore, we conclude that these dislocations **CANNOT** be the dissociated type of $1/3\langle 111 \rangle$ TB disconnections; and they are obviously not the non-dissociated type of $1/3\langle 111 \rangle$ TB disconnections because the extra half $\{111\}$ plane is not on the TB.

Fig. R2 (a) An HRTEM image in this work, showing a TB with steps (marked by the red kink line) as well as a dislocation array on the left side of this TB. (b) An HRTEM image of a dissociated $1/3\langle 111 \rangle$ TB disconnection after Marquis and Medlin¹ (same as **Fig. R1b**). The extra half $\{111\}$ plane (marked by the pink dashed line) of a dissociated $1/3\langle 111 \rangle$ TB disconnection must be located in the region enclosed by the inclined $\{111\}$ plane (marked by the yellow solid line) and the $\{111\}$ TB (marked by the black solid line) that form an **OBTUSE** angle ($\sim 110^\circ$) indicated by the red arrow.

To address the reviewer's concern on our Burgers circuit analysis, we note that **Fig. R1c-g** show several dislocations on the left side of the TB. These dislocations are in the grain rather than on the TB. Hence, the Burgers circuit can be applied to determine the Burgers vector as $1/2\langle 110 \rangle\{111\}$. A clear example of the Burgers circuit is given for dislocation "5" in **Fig. R1f**. From **Fig. R1f** to **Fig. R1g**, the dislocation "5" climbed one atomic spacing upward and did not exhibit any dissociated state on the TB.

In summary, we have provided a thorough analysis of dislocations near the TB and compared our results with the HRTEM images of $1/3\langle 111 \rangle$ TB disconnections by Medlin, Foiles and coworkers¹⁻³. We show that the majority of $1/2\langle 110 \rangle\{111\}$ dislocations near the TB in our work do not lead to the formation of TB steps and thus are neither the non-dissociated nor dissociated $1/3\langle 111 \rangle$ TB

disconnections. For the cases where TB steps are present, the dislocations near the TB cannot be $1/3\langle 111 \rangle$ TB disconnections either. Hence, our work represents the first in situ observations of dislocation climb, leading to the transformation of a high-angle TB to a TB.

Finally, we note that in our recent study not included in this manuscript, dislocation climb was also observed at a low-angle GB in Au (**Fig R3a**). An array of $1/2\langle 110 \rangle\{111\}$ dislocations showed the correlated climbing behavior at room temperature (**Fig R3b-g**). To clarify the climb paths of these GB dislocations, the position of each dislocation in a previous image is overlaid on a subsequent image for tracking dislocation climb trajectories. This result reinforces the notion that dislocation climb in FCC Au does not necessarily need the assistance of a TB or a dissociated core structure. Nonetheless, we greatly appreciate the reviewer's broad knowledge and critical comments on TB disconnections. We have revised the manuscript by noting that the climbing dislocation studied are not $1/3\langle 111 \rangle$ TB disconnections.

Fig. R3 (a) HRTEM image of a low-angle GB. GB dislocations are marked by white symbols and numbers. (b)-(g) Sequential inverse fast Fourier filtered HRTEM images showing the correlated climb of dislocations at this low-angle GB. Dislocations in each image are marked by symbols \perp . The position of each dislocation in a previous image is overlaid on a subsequent image for tracking dislocation climb trajectories.

The authors have also misrepresented and misapplied the Frank-Bilby equation which, as presented here, does not adequately account for the asymmetric inclination of the interface.

Response: Thanks for the critical comment. As noted above, the high-angle GB in this work is the combination of a dislocation array and a TB. Hence, the total misorientation angle between the adjoining grains is the sum of the misorientation associated with the dislocation array and the misorientation associated with the TB. In the original submission, the angle θ in the Frank-Bilby equation is defined as the angle between the respective (111) plane in the two grains (i.e., $(111)_L$ and $(111)_R$ in **Fig. 1a**). As a result, θ changes with climb of dislocations in the array and can be used to track the *change of misorientation* between the two grains due to dislocation climb. When θ is decreased to zero, both $(111)_L$ and $(111)_R$ planes become the

(111) TB plane. Hence, the definition of θ angle is sound and can be used in the Frank-Bilby equation. We have revised the related discussion about θ in the revised manuscript.

While the work presents some nice in situ observations, it is limited by its flawed analysis. Moreover, even if properly analyzed, I don't see that the present results provide sufficient novelty or fundamental new scientific insight to merit publication in Nature Communications.

Response: We greatly appreciate the reviewer's broad knowledge and critical comments on TB disconnections. We hope the above detailed analysis has provided convincing evidence on the dislocation type in our experiment; namely, they are not $1/3\langle 111 \rangle$ TB disconnections. In this work, we highlight the non-negligible role of dislocation climb in the deformation of nanostructured metals at room temperature and further reveal the atomic-scale mechanism of dislocation core climb, at a level not previously possible. Our results demonstrate that dislocation climb under high stresses and at room temperature can induce GB transformation and thus provide a new mechanism of fast GB deformation and evolution. Our findings may offer new insights into the unique time-dependent properties of nanocrystalline materials, such as room-temperature creep, grain coarsening, and among others.

Reviewer #2

Although dislocation climb is a fundamental motion that plays critical roles in the mechanical and plastic properties of materials, knowledge of the atomic-scale mechanisms of dislocation climb remains limited. In this manuscript, the authors reported in situ atomic-scale observations of dislocation climb, which has not been reported previously in the literature. They found that climb of a grain boundary dislocation occurs by reconstruction of two atomic columns at the dislocation core. The new climb mechanism was validated by Monte Carlo simulations. This research is original. The manuscript is well-written. I recommend it for publication after the following comment is addressed.

The reported experiments depend on the free surfaces and applied bending load. Are the strengths of these point defect source and driving force comparable with those in the dislocation climb processes in a wide range of applications?

Response: We thank the reviewer for his/her positive comments on our work. Our finding of room-temperature dislocation climb relies on the high stresses applied as well as on the abundant vacancy sources from both free surfaces and grain boundaries in nanostructured metals. Such conditions are generally applicable to plastically-deforming nanocrystalline materials with plenty of grain boundaries and grain boundary dislocations. Our results offer new insights into the time-dependent plastic behavior of nanocrystalline materials, such as room-temperature creep and grain coarsening. The reported grain boundary transformation mechanisms may not be significant enough in the room-temperature plastic deformation of conventional coarse-grained metals with low flow stresses and a limited fraction of grain boundaries.

Reviewer #3

The authors report in situ HRTEM observations of dislocation climb in a high angle grain boundary (HAGB) in nanoporous gold (np-Au). They find that dislocation climb involves rearrangement of two atomic columns rather than the destruction or construction of a single atomic column, as widely assumed in previous models. They have also performed Grand Canonical Monte Carlo (GCMC) simulations to understand the atomic processes involved in the dislocation climb. The GCMC simulations indicate that merging of two atomic columns into one (for positive climb) and splitting of a single atomic column into two (for negative climb) is energetically more favorable, which supports the in situ HRTEM observations. Although these results are interesting, the main conclusion of the paper, namely, that two atomic columns are involved in the climb process in FCC metals is somewhat premature.

Response: We thank the reviewer for his/her positive comments on our work.

Firstly, the authors assert that there is negligible core dissociation despite the low stacking fault energy of Au. While it is true that no apparent stacking faults are visible in the images, the resolution of the images is not sufficient to conclude that the dislocation core is restricted to a single atomic column. In fact, based on the intensity distribution in some of the HRTEM images (e.g., Supplementary Fig.6a) one might argue that the core is spread over two atomic columns. In that case, it is not surprising to see that two atomic columns are involved in the dislocation climb process.

Response: To address these valuable comments, we carefully examined whether the dislocation cores have dissociated by extracting the corresponding intensity profiles from the HRTEM image at 0 s. As shown in **Fig. R4a**, the ends of extra half-planes are marked by red dots, and they represent dislocation cores. All the intensity profiles at these dislocation cores are extracted from a rectangle area enclosing a {111} atomic layer across each dislocation core (e.g., a yellow dashed region). The extracted intensity profiles of dislocations “1” - “9” are shown in **Fig. R4b**, where the respective intensity valley of each dislocation core is indicated by an arrow. If a dislocation core was to spread over two atomic columns, the intensity distributions for these two atomic columns would increase significantly, thus exhibiting an intensity distribution different from that associated with other neighboring atomic columns. In fact, for most dislocations, the intensity distribution of the atomic column at the end of the extra half-plane is very similar to that associated with the atomic columns on its either side (as indicated by dashed lines in **Fig. R4b**). We notice an exception for dislocation “1”, showing a simultaneous increase in the intensity distribution of two atomic columns at the dislocation core. This implies some degree of dislocation core spreading over the two atomic columns, while the other eight dislocations do not exhibit such kind of core spreading. Hence, core spreading does not appear to have a significant impact on the engagement of two atomic columns in the dislocation climb process.

Fig. R4 (a) HRTEM image of the grain boundary dislocations at 0.0 s. The ends of the extra half-planes are marked by red dots. (b) The corresponding intensity profiles of dislocation “1” – “9” extracted from a rectangle area enclosing a {111} atomic layer across the dislocation cores.

Secondly, and more importantly, the authors do not consider the boundary conditions in interpreting the results. There is a σ_3 boundary on the left of the HAGB whereas there is no GB in the vicinity on the right. Therefore, the stress field near the dislocation core is not symmetric, which might lead to preferential diffusion in one direction and result in the involvement of two atomic columns. When the boundary conditions are symmetric, it is more plausible that three atomic columns (the dislocation core and one column to the left and right) rearrange to form two columns, which would also result in positive climb. While it might be difficult to find a HAGB flanked by two identical GBs on either side in experimental specimens, it should be relatively straightforward to simulate it using GCMC simulations. The authors should simulate this case and verify if only two atomic columns are still involved in the dislocation climb process.

Response: The core structure of a $1/2\langle 011 \rangle$ dislocation in an FCC lattice is intrinsically asymmetric when viewed along the out-of-plane [011] direction. Hence, there is no obvious need for engaging three atomic columns in the dislocation core during dislocation climb. As suggested by the reviewer, we have performed additional GCMC simulations of GB dislocation climb in a symmetric bicrystal under an applied bending load (**Fig. R5a**) using the same scheme described in the manuscript. We still observed two atomic columns are involved in each dislocation climb process. **Fig. R5b1-b4** and **c1-c4** show two representative examples of merging of two atomic columns into one at the core of a climbing dislocation at the GB shown in **Fig. R5a**. Hence, both HRTEM observations and GCMC simulations indicate the preferred process of merging of two atomic columns into one at the core of a climbing GB dislocation.

Fig. R5 (a) GCMC simulation of dislocation climb at a symmetric GB under a symmetric bending load (indicated by arrows). The top view shows the x-y section of the bicrystal. (b1-b4) One example of GCMC snapshots showing the merging of two atomic columns into one at the core of a climbing GB dislocation. The side view shows the x-z section of the bicrystal. (c1-c4) Another example of GCMC snapshots showing the merging of two atomic columns into one at the core of a climbing GB dislocation. The two merging atomic columns are colored by red and blue, respectively, for guiding eyes. The side view shows the x-z section of the bicrystal.

Another aspect that the authors should consider is the possible role of the e-beam in activating the climb process. Although the authors argue that temperature increase due to e-beam exposure is minimal in Au, there are multiple reports of e-beam induced dislocation activation in nanostructured metals (including Au) even in the absence of significant temperature increase. Examples include

1. R. Sarkar, C. Rentenberger, J. Rajagopalan, Electron Beam Induced Artifacts During in situ TEM Deformation of Nanostructured Metals. *Scientific Reports*. 5, 16345 (2015).
2. M. Gaumé, P. Baldo, F. Momprou, F. Onimus, In-situ observation of an irradiation creep deformation mechanism in zirconium alloys. *Scripta Materialia*. 154, 87–91 (2018).
3. S.-H. Li, W.-Z. Han, Z.-W. Shan, Deformation of small-volume Al-4Cu alloy under electron beam irradiation. *Acta Materialia*. 141, 183–192 (2017).
4. S. Stangebye, Y. Zhang, S. Gupta, T. Zhu, O. Pierron, J. Kacher, Understanding and quantifying electron beam effects during in situ TEM nanomechanical tensile testing on metal thin films. *Acta Materialia*, 117441 (2021).

The authors should at least acknowledge this possibility and include the relevant references.

Response: We thank the reviewer for his/her thoughtful comments. We have added the following discussion in the revised manuscript and also cited the suggested references. “Radiolysis to the Au specimen from inelastic scattering of the electron beam is significantly suppressed⁴. Unlike dislocation depinning caused by a displacement cascade effect under ion irradiation⁵, a dislocation cascade induced by electron irradiation is not expected in an Au specimen with high knock-on and sputtering energies⁶. In other words, at an accelerating voltage of 200 keV, knock-on displacements

induced by the electron beam should have a negligible effect on the Au specimen. Therefore, we conclude that the massive vacancy-atom exchange required for dislocation climb in the sample was driven predominantly by high local stresses rather than electron beam irradiation. According to our GPA results and MC simulations, the local normal stress on the HAGB is estimated to be ~3.2 GPa using Young's modulus of ~80 GPa for Au. In addition, in the absence of a native oxide layer on the Au specimen, electron beam-enhanced surface dislocation nucleation was not observed⁷. It has been reported that electron beam irradiation can accelerate a deformation mechanism rather than change to a different mechanism⁸. Hence, electron beam-assisted dislocation activation is possible⁶ and may facilitate the climb and slip behavior of dislocations.”

References

1. Marquis, E. A. & Medlin, D. L. Structural duality of $1/3\langle 111 \rangle$ twin-boundary disconnections. *Philos. Mag. Lett.* **85**, 387-394 (2005).
2. Medlin, D. L., Carter, C. B., Angelo, J. E. & Mills, M. J. Climb and glide of $a/3 \langle 111 \rangle$ dislocations in an aluminium $\Sigma = 3$ boundary. *Philos. Mag. A* **75**, 733-747 (1997).
3. Foiles, S. M. & Medlin, D. L. Structure and climb of $1/3 \langle 111 \rangle$ twin dislocations in aluminum. *Mater. Sci. Eng., A* **319-321**, 102-106 (2001).
4. Egerton, R. F., Li, P. & Malac, M. Radiation damage in the TEM and SEM. *Micron* **35**, 399-409 (2004).
5. Gaumé, M., Baldo, P., Momprou, F. & Onimus, F. In-situ observation of an irradiation creep deformation mechanism in zirconium alloys. *Scripta Mater.* **154**, 87-91 (2018).
6. Sarkar, R., Rentenberger, C. & Rajagopalan, J. Electron Beam Induced Artifacts During in situ TEM Deformation of Nanostructured Metals. *Sci. Rep.* **5**, 16345 (2015).
7. Li, S.-H., Han, W.-Z. & Shan, Z.-W. Deformation of small-volume Al-4Cu alloy under electron beam irradiation. *Acta Mater.* **141**, 183-192 (2017).
8. Stangebye, S., *et al.* Understanding and quantifying electron beam effects during in situ TEM nanomechanical tensile testing on metal thin films. *Acta Mater.*, 117441 (2021).

REVIEWER COMMENTS

Reviewer #1 (Remarks to the Author):

The authors have conducted in situ electron microscopic observations and atomistic modeling of dislocation climb in a nanoscale thin film gold ligament. A novel aspect of the work is the application of a double-tilt straining holder (allowing tilting of the specimen to a suitable zone axis for imaging prior to application of strain). The other novel aspect of this work is the application of a bending mode to the Au ligaments, which provides an elastic driving force to induce dislocation climb.

In my review of the originally submitted manuscript, I expressed concern regarding the interpretation of the observed dislocations. After reading the authors' response, I agree with them now that the dislocations they have observed are not $(1/3)\langle 111 \rangle$ dislocations as I had incorrectly assumed.

Given the novelty of the in situ bending studies, I do believe that this paper could be worthy of publication as a methodological study. That said, I still have several concerns concerning the technical quality of the analysis and with the novelty of the scientific findings. Given these concerns I still find it difficult to recommend this manuscript for publication in its current form without a thorough and complete rewrite and more careful consideration of the technical analysis. Amongst the issues:

(1) The abstract emphasizes the following point: "Our in situ observations also reveal GB transformation through dislocation climb, which suggests a means of controlling microstructures and properties of nanostructured metals." This comment is presumably in reference to the observation of the change in misorientation produced by climb of the dislocation array at the twin boundary (the so-called HAGB that transforms to a twin boundary).

It is well known that dislocations at grain boundaries can climb and moreover that any change in the dislocation content of a grain boundary will necessarily change the character of the boundary. Hence, I do not see the novelty or significance of the authors' observation here.

Moreover, if we accept the authors' interpretation that the HAGB is actually composed of a twin plus a very closely spaced LAGB composed of individual $(1/2)\langle 110 \rangle$ type dislocations, then the so-called "transformation" is simply the removal of the $(1/2)\langle 110 \rangle$ dislocations by climb, which is neither new nor surprising.

(2) Dislocations: The authors argue that the observed dislocations all have identical Burgers vector, $b=(1/2)[0\ 1\ 1]$.

For the given crystallographic geometry, with a line direction parallel with with zone axis $[-1\ 1\ 0]$, such dislocations cannot be in a pure edge configuration. Specifically, $b=(1/2)[011]$ makes a 60° angle with $[-1$

1 0] (i.e., it is mixed dislocation with both edge and screw components). This distinction is important, since if all the dislocation Burgers vectors were identical, the boundary would have an uncompensated screw component and hence a tendency to relax by rotating about the plane of the interface. It is likely, that rather than having identical Burgers vectors, the dislocations, if indeed of type $(1/2)\langle 110 \rangle$, alternate between $(1/2)[011]$ and $(1/2)[101]$, since these have identical edge components, but alternating screw components (which would cancel out). In an HREM image, of course, it is not possible to detect the screw component since it is aligned along the projection direction; thus only the edge components of the Burgers vector are distinguishable. However, determining the full Burgers vectors and the orientations should be easy to determine in the atomistic model since the full 3-D atomic arrangement is available. The authors should provide this analysis.

(3) Line 96: The authors state that the measured misorientation "obeys the Frank-Bilby equation $\theta = 2 \arcsin(b/2d)$..." citing Hirth and Lothe (Second Edition).

(a) The authors need to quantitatively support this assertion by showing that the actual distribution of dislocations is indeed consistent with the measured orientations. Here they should be precise with respect to the actual components of the Burgers vectors (e.g., if indeed the dislocations are $(1/2)\langle 110 \rangle$ then proper account needs to be made of the edge and screw components of these dislocations as noted above.

(b) In particular, the authors have used $b = 0.288$ nm in their calculations. This is the absolute magnitude of a dislocation with $b = (a/2)\langle 110 \rangle$. However, as noted above, given the geometry provided by the authors, the $(a/2)\langle 110 \rangle$ dislocations must be in a mixed orientation, i.e., with b at 60 degrees with respect to the line direction. In terms of accounting for misorientation about the $[-1\ 1\ 0]$ axis, the edge component b should be applied (i.e., for a 60 degree dislocation, $|(a/4)\langle 112 \rangle| = a * \sqrt{6}/4 = 0.25$ nm.

(c) Finally, as a note of precision, the expression they have cited is not the Frank-Bilby equation, per se; rather, this is a simplified expression limited to a symmetric dislocation tilt wall. In the experimental observations neither the dislocation wall (nor the combination of the dislocation wall with the twin boundary) have symmetric inclinations. A proper application of the Frank-Bilby equation would relate the measured dislocation distribution to both the inclination and misorientation of the interface. That said, I appreciate that the expression employed by the authors may still give a reasonable approximation.

(4) In my view, the authors are too cavalier in dismissing the potential for electron beam induced effects. While I agree that at 200 keV the electron energy is below the knock-on displacement threshold in Au, the electron beam can still transfer significant energy to the individual Au atoms (up to 2.67 eV for 200 keV electrons).

(5) The authors claim that the dislocations are not dissociated. I agree with their assessment that extensive dissociation is likely constrained due to the nearby twin boundaries; however, it is unclear from the provided images whether the dislocation cores are dissociated on the scale of a few atom spacings. This is important, since it does not seem at all surprising that climb of dislocations with even narrowly dissociated cores should occur through atom re-arrangements on a similar scale. Indeed, the authors should review and discuss the literature on climb of dissociated dislocations more thoroughly.

(6) Much of the interpretation rests on the strain-mapping results provided in the supplementary materials. The authors provide a citation to the original methodological work on the GPA strain mapping method on line 99 (reference 33, Hýtch et al.). However, the GPA method is extremely sensitive to the exact parameters employed in the mapping, including the choice of g-vectors, the choice of reference frame, and the virtual aperture size and character (e.g., whether gaussian smoothing is employed). Additionally, applying the GPA method to strain fields at an interface is non-trivial since the results are very sensitive to which crystal is chosen as the reference and it can be ambiguous how to deal with the strains at the interface due to the discontinuous change in crystal frame. The authors need to provide the specific details on how they have done the GPA strain mapping and also discuss the sensitivity of their results to choices of the analysis parameters.

(7) The authors draw much significance regarding the proposed dislocation core reconstructions from subtle changes in the observed experimental HRTEM image contrast. I am very skeptical about this interpretation.

(a) Very small changes in crystal orientation (e.g., on the order of mrad) can have huge effects on image contrast in HRTEM. It seems difficult to imagine that such precision can be maintained given the complex dynamical changes occurring during any in situ straining experiment (even with a double-tilt straining holder).

(b) The asymmetry and streaking in the images and variations in contrast across the different grains suggests further that the grains are not exactly aligned (I appreciate the experimental difficulty of such precise alignment -- the point is that in such cases one must be very cautious in interpretation of contrast features).

(c) The authors base their interpretation of the image intensities and contrast changes on image simulations as discussed in the supplementary materials. The problem is that the image simulations provided in the supplementary material have been done using a spherical aberration coefficient (C_s) of 0.5 mm, which is a typical value for an un-corrected system, but as discussed in the methods section (starting at line 373) the microscopy was conducted using aberration corrected lens optics, which the authors indicate had $C_s = -1.15$ microns. The image contrast with such differing aberration coefficients will be very different. (Incidentally, the authors quote the measured C_s value, but neglect to list any of the other higher aberration coefficients, which would presumably have been measured by the CEOS

corrector software along with the measurement of Cs.)

(d) Another potential concern is the image simulation cells themselves. The authors have chosen to construct these cells with free surfaces surrounding all the interfaces. The problem with this approach is that the extensive delocalized Fresnel contrast due to the different vacuum/crystal interfaces potentially interferes with the details of the simulated contrast. While this approach is reasonable perhaps for the top and bottom of the cells (i.e., in the y-direction), since this would mimic the geometry of the ligaments, it does not seem realistic for the sides of the cells (i.e., in the x-direction). An alternative approach would be to employ a periodic cell in the

Given the above problems, I find it difficult to draw any meaningful conclusions from the subtle contrast changes in the dislocation core regions.

Reviewer #2 (Remarks to the Author):

I recommend this revised manuscript for publication.

Reviewer #3 (Remarks to the Author):

The authors have adequately addressed my comments and questions. In particular, the GCMC simulations of the symmetric GB under bending showing that two atomic columns are involved in the climb process is quite convincing.

Point-by-point response to reviewer's comments (NCOMMS-21-29277A-Z)

We sincerely thank the reviewer for his/her careful reading of our manuscript and constructive comments on our work. In the following, our point-by-point response to each comment is highlighted in blue. We have revised the manuscript (highlighted in red) and supplementary materials accordingly.

Reviewer #1:

The authors have conducted in situ electron microscopic observations and atomistic modeling of dislocation climb in a nanoscale thin film gold ligament. A novel aspect of the work is the application of a double-tilt straining holder (allowing tilting of the specimen to a suitable zone axis for imaging prior to application of strain). The other novel aspect of this work is the application of a bending mode to the Au ligaments, which provides an elastic driving force to induce dislocation climb.

In my review of the originally submitted manuscript, I expressed concern regarding the interpretation of the observed dislocations. After reading the authors' response, I agree with them now that the dislocations they have observed are not $(1/3)\langle 111 \rangle$ dislocations as I had incorrectly assumed.

Given the novelty of the in situ bending studies, I do believe that this paper could be worthy of publication as a methodological study. That said, I still have several concerns concerning the technical quality of the analysis and with the novelty of the scientific findings. Given these concerns I still find it difficult to recommend this manuscript for publication in its current form without a thorough and complete rewrite and more careful consideration of the technical analysis. Amongst the issues:

Response: We thank the reviewer for these valuable technical comments. Accordingly, we have thoroughly revised the manuscript. We hope the changes made to the revised manuscript can completely placate the reviewer's concerns and satisfy the requirements for publication in Nature Communications.

(1) The abstract emphasizes the following point: "Our in situ observations also reveal GB transformation through dislocation climb, which suggests a means of controlling

microstructures and properties of nanostructured metals." This comment is presumably in reference to the observation of the change in misorientation produced by climb of the dislocation array at the twin boundary (the so-called HAGB that transforms to a twin boundary).

It is well known that dislocations at grain boundaries can climb and moreover that any change in the dislocation content of a grain boundary will necessarily change the character of the boundary. Hence, I do not see the novelty or significance of the authors' observation here.

Moreover, if we accept the authors' interpretation that the HAGB is actually composed of a twin plus a very closely spaced LAGB composed of individual $(1/2)\langle 110 \rangle$ type dislocations, then the so-called "transformation" is simply the removal of the $(1/2)\langle 110 \rangle$ dislocations by climb, which is neither new nor surprising.

Response: We thank the reviewer for his/her positive comments on our *in situ* TEM technique and loading mode. A key novelty of this work is the stress-driven climb of GB dislocations at **room temperature**. In comparison with dislocation glide, the works on dislocation climb, in particular at room temperature, are rather scarce because dislocation climb generally occurs at elevated temperatures at which the resolution of TEM is limited by thermal drift. In fact, in modern thin-film systems for micro-electronics or micro-electro-mechanical systems, dislocation climb could contribute to plastic flow to a significant extent even at room temperature, e.g., in metallic interfaces^{1,2}. To our knowledge, despite the importance of dislocation climb in nanostructured materials, *in situ* atomic-scale experimental observations of dislocation climb have not been realized until now. In this work, we, for the first time, captured a direct, dynamic, atomic-scale process of GB dislocation climb driven by applying a bending load to nanostructured Au at room temperature. The resulting GB transformation through dislocation climb at **room temperature** has never been reported before, and hence is new, innovative, and surprising. The prevailing high stresses in the nanostructured sample greatly promote the occurrence of GB dislocation climb at room temperature. Our combined HRTEM analysis and atomistic simulations

further reveal the evolution of GB dislocation cores during the climb process, which is also the first of its kind.

We understand the reviewer's concern on the usage of "GB transformation". To avoid possible confusion, we have revised the usage as "GB evolution" in the revised manuscript. In the abstract, to highlight the importance of room temperature creep, we changed the sentence to "*GB evolution through dislocation climb at room temperature*".

(2) Dislocations: The authors argue that the observed dislocations all have identical Burgers vector, $b=(1/2)[0\ 1\ 1]$.

For the given crystallographic geometry, with a line direction parallel with zone axis $[-1\ 1\ 0]$, such dislocations cannot be in a pure edge configuration. Specifically, $b=(1/2)[011]$ makes a 60° angle with $[-1\ 1\ 0]$ (i.e., it is mixed dislocation with both edge and screw components).

This distinction is important, since if all the dislocation Burgers vectors were identical, the boundary would have an uncompensated screw component and hence a tendency to relax by rotating about the plane of the interface. It is likely, that rather than having identical Burgers vectors, the dislocations, if indeed of type $(1/2)\langle 110 \rangle$, alternate between $(1/2)[011]$ and $(1/2)[101]$, since these have identical edge components, but alternating screw components (which would cancel out). In an HREM image, of course, it is not possible to detect the screw component since it is aligned along the projection direction; thus only the edge components of the Burgers vector are distinguishable.

However, determining the full Burgers vectors and the orientations should be easy to determine in the atomistic model since the full 3-D atomic arrangement is available. The authors should provide this analysis.

Response: We thank the reviewer for his/her valuable advice. In response to the reviewer's suggestion, we determined the full Burgers vectors of the dislocations in a 3D atomistic model. We have added the following discussion in the revised supplementary materials.

"The full Burgers vectors and the orientations of the dislocations were determined

through an atomistic model, as shown in **Supplementary Fig. 2 (Fig. R1 in this Response)**. **Fig. R1a** presents the atomistic model of the Au ligament reconstructed from the HRTEM image of **Fig. 1a** in the manuscript, where two neighboring dislocation cores are colored and boxed in black. **Fig. R1b** presents the magnified view of the two dislocations. The red and blue atoms represent two alternating ($\bar{1}10$) layers perpendicular to the zone axis. From the Burgers circuit analysis (non-closure black lines), the Burgers vectors of the two dislocations are marked by the yellow arrows, respectively. Note that the yellow Burgers vector of the top dislocation points from the red to blue atom, while that of the bottom dislocation from the blue to red. This indicates that the out-of-plane screw components of the two Burgers vectors have the opposite signs and they are determined as $1/4[\bar{1}\bar{1}0]$ and $1/4[\bar{1}10]$, respectively. Thus, the full Burgers vectors of these dislocations can be identified as $1/2[011](11\bar{1})$ or $1/2[101](11\bar{1})$.”

Fig. R1 (a) Atomistic model of the Au ligament reconstructed from the HRTEM image in **Fig. 1a**. Atoms in the two neighboring dislocation cores are colored and boxed. **(b)** Magnified image of the two dislocation cores boxed in (a). The red and blue atoms represent two alternating ($\bar{1}10$) layers perpendicular to the zone axis. From the Burgers circuit analysis (non-closure black lines), the Burgers vectors of the two dislocations are marked by the yellow arrows, respectively.

(3) Line 96: The authors state that the measured misorientation "obeys the Frank-Bilby equation $\theta=2 \arcsin (b/2d)$..." citing Hirth and Lothe (Second Edition).

(a) The authors need to quantitatively support this assertion by showing that the actual distribution of dislocations is indeed consistent with the measured orientations. Here they should be precise with respect to the actual components of the Burgers vectors (e.g., if indeed the dislocations are $(1/2)\langle 110 \rangle$ then proper account needs to be made of the edge and screw components of these dislocations as noted above.

(b) In particular, the authors have used $b=0.288$ nm in their calculations. This is the absolute magnitude of a dislocation with $b=(a/2)\langle 110 \rangle$.

However, as noted above, given the geometry provided by the authors, the $(a/2)\langle 110 \rangle$ dislocations must be in a mixed orientation, i.e., with b at 60 degrees with respect to the line direction. In terms of accounting for misorientation about the $[-1\ 1\ 0]$ axis, the edge component b should be applied (i.e., for a 60 degree dislocation, $|(a/4)\langle 112 \rangle| = a * \sqrt{6}/4 = 0.25$ nm.

(c) Finally, as a note of precision, the expression they have cited is not the Frank-Bilby equation, per se; rather, this is a simplified expression limited to a symmetric dislocation tilt wall. In the experimental observations neither the dislocation wall (nor the combination of the dislocation wall with the twin boundary) have symmetric inclinations. A proper application of the Frank-Bilby equation would relate the measured dislocation distribution to both the inclination and misorientation of the interface. That said, I appreciate that the expression employed by the authors may still give a reasonable approximation.

Response: We thank the reviewer for his/her critical comments on the application of the Frank-Bilby equation. As pointed out by the reviewer, the edge component of the Burgers vector, i.e., $b = 0.25$ nm should be used in the quantitative analysis of the dislocation distributions and the misorientation angle. On the other hand, as mentioned in the manuscript, since the HAGB actually consists of a TB and an array of dislocations aligned vertically above each other, the angle θ between the respective (111) plane of the two grains (i.e., $(111)_L$ and $(111)_R$ in **Fig. 1a**) results from the dislocation array and can be measured to track the change of misorientation between the two grains. After clarifying these two points, we will explain in detail the application of the Frank-Bilby

equation to our observations. Indeed, the relation between θ and the average dislocation spacing at the GB in our previous submission is not the Frank-Bilby equation per se, and it is a simplified expression for a symmetric dislocation tilt wall. Hence, we have removed this relation in the revised manuscript. We have added the following discussion on the application of the Frank-Bilby equation in the revised supplementary materials. The Frank-Bilby equation^{3,4} can be written as

$$\mathbf{B}(\mathbf{P}) = (\mathbf{S}_\beta^{-1} - \mathbf{S}_\alpha^{-1})\mathbf{P}$$

where \mathbf{B} is the total Burgers vector intersected by a probe vector \mathbf{P} , and \mathbf{S}_α^{-1} , \mathbf{S}_β^{-1} respectively are inverse matrices of the distortion transformation matrices \mathbf{S}_α , \mathbf{S}_β that map the lattice vectors from the natural unstrained lattices α , β to the reference lattice.

One can solve the above Frank-Bilby equation by selecting a reference state and defining the corresponding probe vector \mathbf{P} . For a given grain boundary with a large angle of θ , the reference crystal can be orientated such that the grain boundary can be produced by rotations $\theta/2$ and $-\theta/2$. In this situation, \mathbf{P} is orthogonal to \mathbf{B} and the equation can be written as⁵

$$\mathbf{B} = 2\sin\frac{\theta}{2}(\mathbf{P} \times \mathbf{a})$$

where \mathbf{a} is a unit vector along the axis of rotation. This equation corresponds to Equation (19-14) on Page. 714 of *Theory of dislocations* (2nd Edition, by Hirth and Lothe⁵) and the probe vector \mathbf{P} here corresponds to the vector \mathbf{V} in Equation (19-14). This equation is also referred as to the Frank's equation in the revised manuscript. According to this equation, a decrease in the number of GB dislocations would lead to decreased \mathbf{B} and consequently decreased GB angle θ . In our experimental observations, GB dislocation climb is responsible for in-plane grain rotation. Thus, \mathbf{P} is perpendicular to \mathbf{a} . The equation can be written as^{3,5}

$$\frac{|\mathbf{B}|}{|\mathbf{P}|} = 2\sin\frac{\theta}{2}$$

Here $|\mathbf{B}|$ can be calculated by multiplying **the edge components** of the GB dislocations (0.25 nm) with the number of GB dislocations. $|\mathbf{P}|$ can be obtained by measuring the total dislocation spacing. The topmost dislocation is not considered for its unknown

dislocation spacing. We can then compare the calculated θ by the equation above with our experimental measurements. For example, at $t = 0$ s, $|\mathbf{B}|$ is calculated to be 2.0 nm and the total dislocation spacing $|\mathbf{P}|$ is 4.8 nm. The calculated θ is 24.0° , which is consistent with the measured θ (24.6°). As shown in **Supplementary Fig. 6b** in the revised supplementary materials (**Fig. R2** in this Response), the calculated θ agrees closely with the measured θ during *in situ* straining. It should be noted that only the edge components of the Burgers vectors which are perpendicular to the tilt GB plane are considered because the screw components which are parallel to the GB plane are undetectable in the HRTEM images and they do not contribute to the tilt angle of the GB. The discussion has been added to the revised supporting information.

Fig. R2 Calculated and measured θ as a function of time, showing close agreement.

(4) In my view, the authors are too cavalier in dismissing the potential for electron beam induced effects. While I agree that at 200 keV the electron energy is below the knock-on displacement threshold in Au, the electron beam can still transfer significant energy to the individual Au atoms (up to 2.67 eV for 200 keV electrons).

Response: We thank the reviewer for his/her concern on the electron beam induced effects. In our previous supplementary materials, only the temperature rise induced by electron beam heating was discussed. In response to this comment, we have added the following discussion on the knock-on effect in the revised supplementary materials.

Under electron-beam irradiation, the maximum energy transferred to a Au atom in a

perfect lattice can be estimated as⁶

$$\Delta E_{\max} = \frac{2E(E + 2m_e c^2)}{Mc^2}$$

Here E is the electron energy, m_e is the electron mass, c is the velocity of light and M is the mass of the Au atom. The maximum energy that an electron beam can transfer to the Au atom is ~ 2.66 eV when the TEM is operated at 200 kV, which is much lower than the threshold displacement energy (i.e., a minimum amount of kinetic energy transferred to a lattice atom that results in the formation of a point defect) of ~ 36 eV for Au⁷. Hence, when the TEM is operated at 200 kV, the knock-on displacement of electron beam on Au atoms is negligible.

The stability of GBs under prolonged electron beam irradiation was also evidenced, as shown in **Supplementary Fig. 19** in the revised supplementary materials (**Fig. R3** in this Response). **Fig. R3** shows the *in situ* HRTEM images of a GB in another Au ligament under electron beam irradiation without applied straining. It is clear that no significant events of dislocation climb, dislocation glide, GB evolution, and rotation take place. These results demonstrate that the GBs and dislocations in Au ligaments under 200 keV electron beam irradiation are stable.

Fig. R3 *In situ* HRTEM images of a GB in an Au ligament under electron beam irradiation without applied straining. The GB is marked by the red line in (a) and the TBs in Grain 2 are marked by the white dashed lines. Both GB and TB show high stability under electron beam irradiation.

(5) The authors claim that the dislocations are not dissociated. I agree with their

assessment that extensive dissociation is likely constrained due to the nearby twin boundaries; however, it is unclear from the provided images whether the dislocation cores are dissociated on the scale of a few atom spacings. This is important, since it does not seem at all surprising that climb of dislocations with even narrowly dissociated cores should occur through atom re-arrangements on a similar scale. Indeed, the authors should review and discuss the literature on climb of dissociated dislocations more thoroughly.

Response: We thank the reviewer for the constructive comments on dislocation dissociation. We acknowledge that a direct combination of HRTEM observations and image simulations is needed to precisely analyze the dissociation core width of dislocations^{8,9}. However, in our *in situ* straining experiments, it is very difficult to perform simulations with such high precision, especially in the vicinity of GB/TB structures under applied stresses. Nevertheless, to address this question, as in our previous response to reviewer #3, we tried to determine whether the cores have dissociated by the variation of the bright-dark contrast at the dislocation cores.

As shown in **Fig. R4a**, the ends of the extra half-planes are marked by red dots, and they represent dislocation cores. All the intensity profiles at these dislocation cores are extracted from a rectangle area enclosing a {111} atomic layer across each dislocation core (e.g., a yellow dashed region). The extracted intensity profiles of dislocations “1” - “9” are shown in **Fig. R4b**, where the respective intensity valley of each dislocation core is indicated by an arrow. If a dislocation core was to spread over two atomic columns, the intensity distributions for these two atomic columns would increase significantly, thus exhibiting an intensity distribution different from that associated with other neighboring atomic columns. In fact, for most dislocations, the intensity distribution of the atomic column at the end of the extra half-plane is very similar to that associated with the atomic columns on either side (as indicated by dashed lines in **Fig. R4b**). We notice an exception for dislocation “1”, showing a simultaneous increase in the intensity distribution of two atomic columns at the dislocation core. This implies some degree of dislocation core spreading over the two atomic columns, while the other

eight dislocations do not exhibit such kind of core spreading. Hence, the dislocation core spreading is not significant and should not affect the engagement of two atomic columns in the dislocation climb process.

Fig. R4 (a) HRTEM image of the grain boundary dislocations at 0 s. The ends of the extra half-planes are marked by red dots. (b) The corresponding intensity profiles of dislocation “1” – “9” extracted from a rectangle area enclosing a {111} atomic layer across the dislocation cores.

In fact, the dislocation core structures, as well as configurations of jogs, have a significant influence on the motion of dislocations^{10,11}. A theory for climb of undissociated edge dislocations has been given by Friedel¹². It has been pointed out that the climb of edge dislocations must occur by the climb of individual jogs along the dislocations by absorption or emission of vacancies. In the case of a dissociated dislocation which consists of two Shockley partial dislocations connected by a stacking fault ribbon, the scenario becomes complicated and even contradictory. Stroh proposed that the jogs on dissociated dislocation lines have first to be constricted before dislocation climb¹³. Non-conservative motion of the constricted jogs along dissociated edge dislocation lines has been observed in materials with low stacking fault energies^{14,15}. However, Thomson and Balluffi argued that the climb of dissociated dislocation can be achieved by nucleation of prismatic vacancy loops at one of the Shockley partial dislocations followed by their propagation across the stacking fault

ribbon toward another partial dislocation, and thus no constriction of jogs is required¹⁰. Grilhé et al. analyzed the model in detail later¹⁶. The climb of dissociated dislocations in such a model was observed experimentally in irradiated alloys or semiconductors¹⁷⁻²⁰ and quenched alloys from high temperature²¹. Later, using a diffusive molecular dynamics method, Sarkar et al. studied the evolution of a jog-pair in FCC Cu and identified a displacive-diffusive path associated with climb²². Although this pathway is distinctly different from the one proposed by Thomson and Balluffi, the evolution of the dislocation lines agrees overall with the Thomson-Balluffi mechanism.

However, it is worth noting that the Thomson-Balluffi mechanism is suitable when high vacancy supersaturations exist. When there is no vacancy supersaturation, Argon and Moffatt suggested that the climb of dissociated dislocations is controlled by vacancy emission from extended jogs²³. Based on a hard sphere model, it was indicated that the climb of an extended acute jog consists of translation of an atom row upward, acquisition of an atom from the surrounding material, and generation of a vacancy that has to be diffused away. Despite these theoretical studies, the detailed mechanisms for dissociated dislocation climb are still poorly understood since the climb process is essentially three-dimensional and involves atomic-scale interactions between point defects and dislocation jogs. Therefore, most of the discussions focus on simplified cases of undissociated dislocations, similar to the case of our experimental observations.

(6) Much of the interpretation rests on the strain-mapping results provided in the supplementary materials. The authors provide a citation to the original methodological work on the GPA strain mapping method on line 99 (reference 33, Hÿtch et al.). However, the GPA method is extremely sensitive to the exact parameters employed in the mapping, including the choice of g-vectors, the choice of reference frame, and the virtual aperture size and character (e.g., whether gaussian smoothing is employed). Additionally, applying the GPA method to strain fields at an interface is non-trivial since the results are very sensitive to which crystal is chosen as the reference and it can be ambiguous how to deal with the strains at the interface due to the discontinuous change

in crystal frame. The authors need to provide the specific details on how they have done the GPA strain mapping and also discuss the sensitivity of their results to choices of the analysis parameters.

Response: We thank the reviewer for the suggestion about the GPA straining mapping analysis. As pointed out by the reviewer, GPA results are sensitive to the choice of parameters in the mapping. Before discussing the effects of the choice of parameters on the results, we note that the GPA method was applied to **the lattice of the right grain** rather than to the interface. In other words, the strain distribution of the lattice **near** the GB was analyzed to investigate the mechanical loading on the ligament.

In accordance with this comment, we provided more details on our GPA analysis and discussed the influences of the choices of **g**-vectors and reference frame on the GPA results in detail. As an example, **Fig. R5a** shows the HRTEM image of the Au ligament at 171.8 s. The HAGB region is marked by a white polygon. **Fig. R5b** and **Fig. R5c** present the FFT image of the ligament, where different sets of **g**-vectors are circled in blue and red, respectively. In **Fig. R5b**, the **g**-vectors of $(11\bar{1})_R$ and $(002)_R$ are chosen and the corresponding stain map of ϵ_{xx} is shown in **Fig. R5d**. In **Fig. R5c**, the **g**-vectors of $(002)_R$ and $(111)_R$ are chosen and the corresponding stain map of ϵ_{xx} is shown in **Fig. R5e**. As denoted by the white squares in **Fig. R5d** and **Fig. R5e**, the same regions are selected as reference frames to compare the influence of the choice of **g**-vectors. The reference frames are far from the GB region and are located in the lower middle of the ligament (i.e., almost on the neutral plane) so that the strain in the reference regions is nearly zero. It is clear that **Fig. R5d** and **Fig. R5e** show similar strain distributions of the right grain. The strain values on the GB plane are measured along the arrows, and the corresponding profiles are shown in **Fig. R5f** and **Fig. R5g**, respectively. Both strain distributions indicate the compressive and tensile lattice strain on the upper and lower part of the GB plane, respectively. According to the strain maps and the profiles of the strain values, the choice of **g**-vectors does not affect the GPA results. However, it is necessary to select the non-parallel **g**-vectors belonging to the right grain.

Fig. R5 Lattice strain maps with different choices of \mathbf{g} -vectors. (a) HRTEM image of the Au ligament. (b-c) FFT images of the ligament. Different sets of \mathbf{g} -vectors are circled in blue and red, respectively. (d-e) Lattice strain maps in (d) and (e) corresponding to the selected \mathbf{g} -vectors in (b) and (c), respectively. (f-g) Profiles of strain values on the GB plane extracted from the rectangles in (d) and (e), respectively.

Fig. R6a and **Fig. R6b** show lattice strain maps of ϵ_{yy} and ϵ_{xy} using the \mathbf{g} -vectors selected in **Fig. R5b**. In general, both strain distributions are uniform in the right grain. Profiles of strain values extracted from the rectangles in **Fig. R6a** and **Fig. R6b** are presented in **Fig. R6c** and **Fig. R6d**, respectively. It can be seen that both values of ϵ_{yy} and ϵ_{xy} are close to zero, ranging from $\sim -0.5\%$ to $\sim 0.5\%$.

Fig. R6 Lattice strain maps of ϵ_{yy} and ϵ_{xy} (a and b) and the corresponding profiles of strain value (c and d).

To discuss the effect of the choice of reference frame on the GPA results, we set the

reference frame in the region that deviates from the original position and compared the changes in the strain distribution. **Fig. R7a** shows the strain map in **Fig. R5d**. As mentioned before, the \mathbf{g} -vectors of $(11\bar{1})_R$ and $(002)_R$ are chosen (also see the inset). Although it may not be a “true” zero-strain region, the reference frame is placed far from the GB region in the lower middle of the ligament, i.e., on the neutral plane, so that the strain within the region is very close to zero. The profile of the strain value extracted from the rectangle is shown below the strain map. The crossover from compressive to tensile strain is marked with a red line. In **Fig. R7b**, the reference frame (marked by the solid line box) is located above the previous position (marked by the dashed box) but still as far away from the GB region as possible. It can be seen from the profile that as the reference frame is positioned away from the neutral plane and may be affected by residual compressive stress, the strain value is generally increased and the crossover from compressive to tensile strain slightly deviates from its original position. Conversely, the reference frame is placed below the previous position in **Fig. R7c**. As shown in the profile, probably affected by residual tensile stress, the strain value decreases and the crossover also deviates from the original position. In summary, the choice of reference frame has a large effect on the absolute value of strain but does not affect the relative value of strain on the GB plane. Importantly, the lattice strain always changes from compression to tension on the GB plane from the top to the bottom, which is consistent with the distribution of the applied bending stress.

Fig. R7 Lattice strain maps with different choices of reference frame and the

corresponding profiles of strain value.

In the above GPA analysis, a Gaussian smoothing parameter of 5.0 was applied. Masks with radii of 1/5 of the corresponding \mathbf{g} -vectors were used for generating the lattice strain maps. The mask size shows the area selected in the Fourier space around the spot of interest and the inverse value of the selected spot corresponds to the effective spatial resolution of the lattice strain maps. It is worth noting that the spatial resolution and the precision are roughly inversely proportional to each other, and a compromise between them must be made to obtain reliable lattice strain maps.

(7) The authors draw much significance regarding the proposed dislocation core reconstructions from subtle changes in the observed experimental HRTEM image contrast. I am very skeptical about this interpretation.

(a) Very small changes in crystal orientation (e.g., on the order of mrad) can have huge effects on image contrast in HRTEM. It seems difficult to imagine that such precision can be maintained given the complex dynamical changes occurring during any in situ straining experiment (even with a double-tilt straining holder).

(b) The asymmetry and streaking in the images and variations in contrast across the different grains suggests further that the grains are not exactly aligned (I appreciate the experimental difficulty of such precise alignment -- the point is that in such cases one must be very cautious in interpretation of contrast features).

(c) The authors base their interpretation of the image intensities and contrast changes on image simulations as discussed in the supplementary materials. The problem is that the image simulations provided in the supplementary material have been done using a spherical aberration coefficient (C_s) of 0.5 mm, which is a typical value for an uncorrected system, but as discussed in the methods section (starting at line 373) the microscopy was conducted using aberration corrected lens optics, which the authors indicate had $C_s = -1.15$ microns. The image contrast with such differing aberration coefficients will be very different. (Incidentally, the authors quote the measured C_s

value, but neglect to list any of the other higher aberration coefficients, which would presumably have been measured by the CEOS corrector software along with the measurement of Cs.)

(d) Another potential concern is the image simulation cells themselves. The authors have chosen to construct these cells with free surfaces surrounding all the interfaces. The problem with this approach is that the extensive delocalized Fresnel contrast due to the different vacuum/crystal interfaces potentially interferes with the details of the simulated contrast. While this approach is reasonable perhaps for the top and bottom of the cells (i.e., in the y-direction), since this would mimic the geometry of the ligaments, it does not seem realistic for the sides of the cells (i.e., in the x-direction). An alternative approach would be to employ a periodic cell in the

Given the above problems, I find it difficult to draw any meaningful conclusions from the subtle contrast changes in the dislocation core regions.

Response: We thank the reviewer for his/her concern on the interpretation of HRTEM image contrast. We appreciate the reviewer for pointing out the inconsistency of the Cs value between the experiments and simulations (the comment (c)). Indeed, a wrong spherical aberration coefficient (Cs) of 0.5 mm was used in our previous HRTEM image simulation. We re-performed HRTEM image simulations with the same Cs value as the experiment and discussed the effects of defocus value, sample thickness, and crystal orientation on the simulation results.

The structure model was reconstructed based on the HRTEM image of the Au ligament at $t = 0$ s. The possible sample thicknesses were set as 8, 16, and 24 layers of (1 $\bar{1}$ 0) atomic plane, which corresponded to thicknesses of approximately 2.4 nm, 4.8 nm, and 7.1 nm, respectively. HRTEM image simulation was conducted using the commercial xHREM software (HREM RESEARCH INC.), which emerges from the image simulation programs based on the FFT multislice technique developed by Ishizuka^{24,25}. **Table R1** presents the detailed parameters used in the image simulations.

Table R1. Parameters used in the HRTEM image simulations.

Acceleration voltage	200 kV
Spherical aberration coefficient	-1.15 μm
Defocus spread	3 nm
Beam convergence	2 mrad
Defocus value (underfocus)	7.5 nm; 10 nm; 12.5nm

Fig. R8 shows a series of simulated HRTEM images of the Au ligament. The sample thicknesses and defocus values are given in the corresponding images. With appropriate parameters, the dark spots represent Au atomic columns while the white spots represent the channels between atomic columns in all simulated images, as indicated by the structure model overlapped with the top left image. This is also evidenced by the simulated phase contrast transfer function (CTF) in **Fig. R9** with the parameters above and a defocus value of 10 nm. Importantly, with a minimized Cs value and appropriate defocus values, the extensive Fresnel contrast in our previous simulation results which was also mentioned by the reviewer in (d) is greatly eliminated.

As can be seen from the magnified images of the white squared regions, the simulated images vary dramatically for different defocus values in an increment of 2.5 nm as well as different sample thicknesses. In the new simulations, by refining all the parameters, the sample thickness of 16 layers of (1 $\bar{1}$ 0) atomic plane (i.e., 4.8 nm) and the defocus value of 10 nm are determined to make the simulated image provide a best fit to the experimental images.

Fig. R8 A series of simulated HRTEM images of the Au ligament. Sample thicknesses and defocus values are given in the corresponding images. The insets show the magnified images of the white squared regions, demonstrating details of the simulated results.

Fig. R9 Simulated phase CTF function (black line) and the envelope function (red line).

For the comments (a) and (b), we appreciate the reviewer for recognizing the difficulty and challenge of our atomic-scale *in situ* TEM experiments. We agree with the reviewer that the crystal orientation (or alignment) can significantly influence the image contrast and, as noted by the reviewer, it is quite difficult to make the grains exactly aligned during *in situ* straining. However, our observations are focused on the contrast changes with very small regions of dislocations and obvious orientation changes are not supposed to take place during our observations. With appropriate HRTEM image simulations, we demonstrated that slight grain misalignment may affect the global contrast of the grains, but is unlikely to lead to local contrast change in the dislocation cores. **Fig. R10a** shows a well-aligned structure model of the Au ligament. As indicated by the coordinate axes, z-axis is parallel to the direction of the electron beam. The corresponding simulated image that exhibits the most satisfactory resemblance to the experimental HRTEM image (sample thickness of 16 layers; defocus value of 10 nm) is displayed in **Fig. R10a'**. The inset shows the magnified image of the white squared region. As extracted from the rectangle enclosing a {111} atomic layer, the line profile of normalized intensity across the dislocation core is shown below the simulated image. Similar to the experimental results in Fig. 3 in our manuscript, the line profile exhibits regular fluctuations. **Fig. R10b** and **Fig. R10c** present the structure models rotating 0.5° about x-axis and 0.5° about y-axis, respectively. The corresponding simulated images are shown in **Fig. R10b'** and **Fig. R10c'**. Due to the small changes in crystal orientation, the defocus value was adjusted accordingly, which is analogous to that in the actual TEM operation. Both line profiles below the simulated images display regular fluctuations, which is similar to that in **Fig. R10a'**. That is to say, the slight misalignment of the Au ligament cannot lead to local contrast variation at the dislocation core. In fact, the contrast change within a range of 2-3 atomic columns is mainly caused by the diffusion of atoms/vacancies at the core.

Fig. R10 Simulated images of the Au ligament with different alignments.

While the interpretation of HRTEM image contrast can be difficult, the small sample thickness and minimized spherical aberration make the interpretation more straightforward on the basis of a charge density projection approximation²⁶. For the phase-contrast HRTEM, the image contrast formation can be described by the phase CTF $U(u)$, which can be written as

$$U(u) = \exp(i \phi(u)) = \exp(i\pi\Delta f\lambda u^2 + 0.5i\pi C_s \lambda^3 u^4)$$

where Δf and C_s correspond to a defocus value and a spherical aberration coefficient of the objective lens, λ is the electron wave length and u is the spatial frequency. For the Cs-corrected TEM, the C_s value is small and can be ignored (i.e., $C_s \sim 0$). The contrast transfer function can be re-written as:

$$U(u) = \exp(i \phi(u)) = \exp(i\pi\Delta f\lambda u^2)$$

When the defocus value Δf is small,

$$U(u) = 1 + i\pi\Delta f\lambda u^2$$

For a thin TEM specimen, the specimen transfer function $f(x, y)$ can be simplified by the weak phase-object approximation:

$$f(x, y) = \exp(i\sigma v(x, y))$$

where the $v(x, y)$ is the electrostatic potential and σ is the interaction constant. Then, the CTF modulated wave function on the back-focus plane of the objective lens is given by:

$$F(x, y) = \exp(i\pi\Delta f\lambda u^2) \cdot FT\{f(x, y)\} = \exp(i\pi\Delta f\lambda u^2) \cdot F(u)$$

By inverse Fourier Transform (FT), the wave function at the image plane of the objective lens becomes:

$$\Psi(x, y) = f(x, y) + i\pi\Delta f\lambda FT\{u^2 \cdot F(u)\}$$

Based on FT differential property, the wave function can be written as:

$$\Psi(x, y) = \left\{ 1 - \frac{\Delta f\lambda\sigma \nabla^2 v(x, y)}{4\pi} + \frac{i\Delta f\lambda\sigma^2}{4\pi} \left[\left(\frac{\partial v(x, y)}{\partial x} \right)^2 + \left(\frac{\partial v(x, y)}{\partial y} \right)^2 \right] \right\} \exp(-i\sigma v(x, y))$$

The intensity distribution on the image plane can be derived from the wave function:

$$I(x, y) = \Psi(x, y)\Psi^*(x, y) = 1 - \frac{\Delta f\lambda\sigma \nabla^2 v(x, y)}{2\pi}$$

The relation between electrostatic potential $v(x, y)$ and projected charge density $\rho(x, y)$ can be described by the Poisson equation:

$$\nabla^2 v(x, y) = -4\pi\rho(x, y)$$

Then,

$$I(x, y) = 1 + 2\Delta f\lambda\sigma\rho(x, y)$$

Consequently, the image contrast $c(x, y)$ is:

$$c(x, y) = 2\Delta f\lambda\sigma\rho(x, y)$$

which is linearly proportional to the charge densities of atoms and molecules on the projection plane. Since the projected charge density function $\rho(x, y)$ is positively correlated with the sample thickness, a positive correlation between the image contrast and the sample thickness (i.e., the number of atoms in an atomic column) can be established when the TEM sample is sufficiently thin. This provides a theoretical basis for our semi-quantitative analysis of HRTEM images relating local contrast change with atom/vacancy diffusion at the dislocation cores. We note that the intensity values measured in our HRTEM images may not be linearly related to the number of the atoms within an individual atomic column due to unavoidable misalignment of the sample or

residual optical lens aberrations.

Based on the correct Cs and defocus value, we re-simulated the HRTEM images during climb of a GB dislocation and updated **Supplementary Fig. 14** in the revised supplementary materials. As can be seen from the normalized contrast intensities along the dotted rectangles, the dislocation climb process is the same as the experimental results.

Fig. R11 (a)-(d) Simulated HRTEM images (left) and corresponding atomic configurations (right) during climb of a GB dislocation. The defocus value is set as 10 nm and the sample has 16 layers of $(1\bar{1}0)$ plane. Dark spots in the simulated HRTEM images represent atomic columns. Red atomic columns are used to represent the dislocation core in the reconstructed atomic configuration (right) in (a). A half amount of atoms are being removed from the two red columns in the dislocation core in (b), followed by the merging process in (c). Two red columns merge into one column in (d), indicating the completion of dislocation climb by one atomic layer. The normalized contrast intensities along the dotted rectangles are plotted in (e), showing the same dislocation climb process as the experimental results.

Additionally, to further visualize the contrast variation at the dislocation cores, we processed the experimental HRTEM images with false color. As circled in black, the red spots corresponding to atomic columns at the dislocation core in **Fig. R12a** became faint in **Fig. R12b** due to the massive atom diffusion away. Accordingly, affected by

atom diffusion and rearrangement, the color of the blue spots corresponding to the channels became lighter simultaneously. In **Fig. R12c** and **Fig. R12d**, two red spots merged into a single one, indicating the completion of dislocation climb.

Fig. R12 Experimental HRTEM images with intensity shown in false color.

Finally, regarding the higher aberration coefficients of the TEM, we list the residual aberrations calculated by the CEOS aberration corrector software in **Table R2**.

Table R2. Residual aberrations calculated by the CEOS aberration corrector software.

Residual aberration	Value
Two-fold astigmatism (A_1)	636.6 μm
Three-fold astigmatism (A_2)	25.4 nm
Axial coma (B_2)	13.1 nm
Spherical aberration coefficient (C_3)	-1.15 μm
Four-fold astigmatism (A_3)	1.1 μm
Star aberration (S_3)	1.3 μm
Five-fold astigmatism (A_4)	29.4 μm

References

1. Wang, J., Hoagland, R. G. & Misra, A. Room-temperature dislocation climb in metallic interfaces. *Appl. Phys. Lett.* **94**, 131910 (2009).

2. Li, N., Wang, J., Huang, J. Y., Misra, A. & Zhang, X. In situ TEM observations of room temperature dislocation climb at interfaces in nanolayered Al/Nb composites. *Scripta Mater.* **63**, 363-366 (2010).
3. Frank, F. C. *A symposium on the plastic deformation of crystalline solids* (Office of Naval Research, Pittsburgh 150, 1950).
4. Bilby, B. A. *Bristol conference report on defects in crystalline materials*. Phys. Soc., London, (1955). 123.
5. Hirth, J. & Lothe, J. *Theory of Dislocations*. 2nd, (Wiley, New York, 1982).
6. Banhart, F. Irradiation effects in carbon nanostructures. *Rep. Prog. Phys.* **62**, 1181-1221 (1999).
7. Vajda, P. Anisotropy of electron radiation damage in metal crystals. *Rev. Mod. Phys.* **49**, 481-521 (1977).
8. Mills, M. J. & Stadelmann, P. A study of the structure of Lomer and 60 dislocations in aluminium using high-resolution transmission electron microscopy. *Philos. Mag. A* **60**, 355-384 (1989).
9. Balk, T. & Hemker, K. High resolution transmission electron microscopy of dislocation core dissociations in gold and iridium. *Philos. Mag. A* **81**, 1507-1531 (2001).
10. Thomson, R. M. & Balluffi, R. W. Kinetic Theory of Dislocation Climb. I. General Models for Edge and Screw Dislocations. *J. Appl. Phys.* **33**, 803-816 (1962).
11. Escaig, B. Emission et absorption de défauts ponctuels par les dislocations dissociées. *Acta Metall.* **11**, 595-610 (1963).
12. Friedel, J. *Dislocations*. (Oxford, New York, 1964).
13. Stroh, A. Constrictions and jogs in extended dislocations. *Proc. Phys. Soc. B* **67**, 427 (1954).
14. Carter, C. B. The influence of jogs on the extension of dislocation nodes. *Philos. Mag. A* **41**, 619-635 (1980).
15. Carter, C. & Ray, I. Observations of constrictions on dissociated dislocation lines in copper alloys. *Philos. Mag.* **29**, 1231-1235 (1974).
16. Grilhé, J., Boisson, M., Seshan, K. & Gaboriaud, E. J. Climb model of extended dislocations in f.c.c. metals. *Philos. Mag.* **36**, 923-930 (1977).
17. Cherns, D., Hirsch, P. B. & Saka, H. Mechanism of climb of dissociated dislocations. *Proc. R. Soc. London A* **371**, 213-234 (1980).
18. Saka, H., Kondo, T. & Kiba, N. Direct observation of non-conservative motion of extended jogs on dissociated dislocations. *Philos. Mag. A* **44**, 1213-1218 (1981).
19. Cherns, D. & Feuillet, G. The mechanism of dislocation climb in GaAs under electron irradiation. *Philos. Mag. A* **51**, 661-674 (1985).
20. Thibault-Desseaux, J., Kirchner, H. O. K. & Putaux, J. L. Climb of dissociated dislocations in silicon. *Philos. Mag. A* **60**, 385-400 (1989).
21. Décamps, B., Cherns, D. & Condat, M. The climb of dissociated dislocations in a quenched Cu-13.43 at.% Al alloy. *Philos. Mag. A* **48**, 123-137 (1983).
22. Sarkar, S., *et al.* Finding activation pathway of coupled displacive-diffusional defect processes in atomistics: Dislocation climb in fcc copper. *Phys. Rev. B* **86**, 014115 (2012).
23. Argon, A. S. & Moffatt, W. C. Climb of extended edge dislocations. *Acta Metall.* **29**, 293-299 (1981).
24. Ishizuka, K. Contrast transfer of crystal images in TEM. *Ultramicroscopy* **5**, 55-65 (1980).
25. Ishizuka, K. Multislice formula for inclined illumination. *Acta Cryst. A* **38**, 773-779 (1982).

26. Lynch, D., Moodie, A. F. & O'keefe, M. n-Beam lattice images. V. The use of the charge-density approximation in the interpretation of lattice images. *Acta Cryst. A* **31**, 300-307 (1975).

REVIEWERS' COMMENTS

Reviewer #1 (Remarks to the Author):

I appreciate the authors' thorough and comprehensive additional work to address the technical concerns I raised on my earlier review. With these revisions I support accepting the manuscript.